# Molecular hydrogen in the N-doped LuH$_3$ system as a possible path to superconductivity

Cesare Tresca [1] ✉, Pietro Maria Forcella [2], Andrea Angeletti[3,4], Luigi Ranalli [3,4], Cesare Franchini [4,5], Michele Reticcioli [4] ✉ & Gianni Profeta[1,2]

The discovery of ambient superconductivity would mark an epochal break-through long-awaited for over a century, potentially ushering in unprecedented scientific and technological advancements. The recent findings on high-temperature superconducting phases in various hydrides under high pressure have ignited optimism, suggesting that the realization of near-ambient superconductivity might be on the horizon. However, the preparation of hydride samples tends to promote the emergence of various metastable phases, marked by a low level of experimental reproducibility. Identifying these phases through theoretical and computational methods entails formidable challenges, often resulting in controversial outcomes. In this paper, we consider N-doped LuH$_3$ as a prototypical complex hydride: By means of machine-learning-accelerated force-field molecular dynamics, we have identified the formation of H$_2$ molecules stabilized at ambient pressure by nitrogen impurities. Importantly, we demonstrate that this molecular phase plays a pivotal role in the emergence of a dynamically stable, low-temperature, experimental-ambient-pressure superconductivity. The potential to stabilize hydrogen in molecular form through chemical doping opens up a novel avenue for investigating disordered phases in hydrides and their transport properties under near-ambient conditions.

Hydrides exhibit high-temperature superconductivity under high-pressure[1–4], giving the perception that the ambient-condition super-conductivity (i.e., high-temperature, low-pressure) could be soon achieved. However, sample preparation leads to metastable structural phases, which hinder experimental reproducibility and prove difficult to characterize through theoretical and computational methods. Such metastable phases have been proposed as key to explain peculiar superconducting phases in phosphorus-hydrides[5], and other different compounds, such as phosphorus under pressure[6], gallium[7], and

barium[8]. In particular, hydrogen complexes, such as molecular hydrogen, may form in hydrides under high pressure and/or in hydrogen-rich samples. These complexes exhibit non-trivial effects on the properties of the host materials, potentially either facilitating the emergence of superconductivity or driving the system into an insulating state[9–18].

The intricate field of hydrides' physics poses challenges and uncertainties, with the added complication of retracted publications that initially claimed near-room temperature superconductivity in

[1]CNR-SPIN c/o Dipartimento di Scienze Fisiche e Chimiche, Università degli Studi dell'Aquila, L'Aquila, Italy. [2]Dipartimento di Scienze Fisiche e Chimiche, Università degli Studi dell'Aquila, L'Aquila, Italy. [3]University of Vienna, Vienna Doctoral School in Physics, Vienna, Austria. [4]Faculty of Physics and Center for Computational Materials Science, University of Vienna, Vienna, Austria. [5]Dipartimento di Fisica e Astronomia, Università di Bologna, Bologna, Italy. ✉ e-mail: cesare.tresca@spin.cnr.it; michele.reticcioli@univie.ac.at

sulfur hydride[19], and near-ambient superconductivity in lutetium hydride[20,21]. The recent, deceiving observation of near-ambient conditions superconductivity in $LuH_{3-\delta}N_\varepsilon$ (reported by the Dias' group)[20,21] has been received by the scientific community with skepticism, but, at the same time, with curiosity, as demonstrated by immediate experimental attempts[22–26] to replicate the synthesis and numerous computational works to rationalize the experimental results[27–40]. However, all attempts to reproduce these results proved unsuccessful, with the only exception of a study conducting resistivity measurements on Dias' samples: Nevertheless, the work has remained unpublished to this day (available only as pre-print), raising doubts about the validity of reported results[41]. Ultimately, the work claiming for ambient-condition superconductivity was retracted upon request of most of the authors, who raised concerns about the integrity of the published data[21].

Computer-aided simulations have proven invaluable in this field[42], pre-emptively predicting new high-pressure superconductors before their experimental discovery. Notably, $SH_3$[43] and $LaH_{10}$[44] stand out as exceptional examples. Given the absence of solid experimental evidence for a near-ambient pressure superconducting phase in hydrides, theory emerges as a viable tool to explore the low-pressure physics of hydrides. However, it is essential to note that several theoretical predictions concerning binary and ternary hydrides[45] have not found experimental confirmation. This discrepancy may be attributed to challenges in accurately accounting for real experimental conditions during crystal growth. As an illustration, numerous studies focusing on LuHN ternary hydride have recently proposed different (metastable or dynamically unstable) structures showing sizable critical temperatures[27,30,31,33,34,36,37], yet these predictions have not been experimentally confirmed to date.

In this work, we show that dynamical and disorder effects are crucial to explore the low energy structures at ambient conditions in hydrides. We propose new metastable phases for N-doped Lu hydride, containing hydrogen in molecular form, stabilized by nitrogen impurities, which leads to the emergence of low-temperature, near ambient-pressure superconductivity.

Our machine-learning-accelerated force-field molecular dynamics (MLFF-MD) is able to disclose the formation of $H_2$ molecules inside the Lu matrix.

These molecular phases are found dynamically stable by Density Functional Theory (DFT) calculations showing the emergence of a finite critical temperature ($T_C \simeq 10K$), partly arising from $H_2$ vibrations as found in molecular metallic hydrogen[46].

Our findings suggest a new route for the exploration of disordered phases in hydrides.

## Results

Figure 1 collects the results as obtained from MLFF-MD simulations, modeling $LuH_3$ using a $4 \times 4 \times 4$ unit cell, with a substitutional N doping on H sites of 12.5% in line with the content reported for the experimental samples[20,21,41] at ambient pressure (no external pressure applied to the system).

We initially conducted a thermalization calculation, with a temperature ramping from very low (<1 K) to high (up to 400 K) values, starting with Lu atoms on *fcc* sites, Fm$\bar{3}$m space-group (hydrogen atoms in tetrahedral and octahedral sites of the *fcc* Lu lattice). Nitrogen was substituted on tetrahedral sites, see also Supplementary Figs. 1, 2 in the Supplementary Information (SI) for the structural model and the complete set of MLFF-MD data. In our simulations, while Lu atoms oscillate around the *fcc* sites as expected[20–22,47–49], H atoms tend to form molecules already at very low temperature: as shown in Fig. 1a, $H_2$ molecules start to form spontaneously at approximately 15 K, till a saturation value of one molecule per N atom is reached.

The system exhibits a high degree of disorder, as found in real samples[5,18,50–52], with the molecules randomly distributed (see the

structural models in Supplementary Fig. 2b, c, the pair correlation function in Supplementary Fig. 3 and the time-evolution trajectories in Supplementary Fig. 4). Although overall the total number of $H_2$ molecules equals the number of nitrogen impurities, we observe a local variation with zero, one or two $H_2$ molecules surrounding each N atom (at an average distance of ~2.5 Å). Interestingly, the average $H_2$ bond length is found to be expanded with respect to the gas phase of about 10% (see Supplementary Fig. 5a in the SI), as observed in the high pressure metallic hydrogen phase[53,54], suggesting a partial occupation of anti-bonding orbitals (as confirmed by the Bader charge analysis in Supplementary Table 1 in the SI), and, possibly, the activation of collective interactions, as already reported in superconducting solid hydrogen[46] or superhydrides[2–4,44,55].

The formation of the $H_2$ molecules lowers the total energy of the system (see Supplementary Fig. 1 in the SI): once formed, the $H_2$ molecules appear extremely robust against dissociation and do not show any tendency to the formation of clathrate-like structures. Starting from the structures explored during the thermalization calculations, we have conducted additional MLFF-MD simulations at a temperature of 100 K, observing no dissociation for the whole MLFF-MD duration of 0.3 ns (Fig. SF6), finding that the number of molecules remains constant to one per N impurity. By fixing the temperature to 300 K (Fig. 1b), we observe that $H_2$ molecules tend to dissociate forming short H-N bonds (-1.0 Å, see the structural model in Fig. 1c), without disappearing completely, even in the long time frames of our molecular dynamics simulations. This happens also at 200 K (see Supplementary Fig. 6).

Importantly, the system explores both the metallic and insulating regimes, strongly depending on the structural phase: In case the sum of $H_2$ molecules and H-N bonds at a given time step equals the number of N atoms, we observe an insulating character; metallic otherwise (see the background color of Fig. 1b, and the corresponding density of states in Supplementary Fig. 7). The Bader charge analysis in Supplementary Table 1 explains this behavior. The $Lu^{+3}$ atom shares 3 electrons that are accommodated on the $H^{-1}$ atoms. Substituting one $H^{-1}$ with the $N^{-3}$ dopant, frees two hydrogen atoms that can now bond independently with each other, forming an $H_2$ molecule (accommodating only a tiny amount of charge from the crystal, 0.2 e). Alternatively, one of the two hydrogen atoms can form an H-N bond, entering a $H^{+1}$ valence state, while the other atom retains its $H^{-1}$ state, keeping the system insulating. Conversely, in the metallic regime, the number of $H_2$ molecules and H-N bonds does not equal the number of N impurities, leaving some electronic charge uncompensated: The Bader charge analysis shows that the excess electrons are hosted on the metallic Lu orbitals.

We find that the formation of the $H_2$ molecules is promoted by the N-substitution.

As a further proof, we performed two additional sets of MLFF-MD calculations for the pristine $LuH_3$ system (0% content of N) in the Fm$\bar{3}$m phase: no $H_2$ molecules have spontaneously formed, at variance with the N-doped systems. Furthermore, starting the simulation with artificially formed $H_2$ molecules in the undoped $LuH_3$ unit cell, the MLFF-MD simulations reveals a clear tendency towards a complete dissociation of all $H_2$ molecules (see Supplementary Fig. 8 in the SI).

The formation of $H_2$ molecules represents a new aspect in the physics of superconducting hydrides, therefore, it is worth to analyze their effects on the electronic and dynamical properties of the representative $LuH_{2.875}N_{0.125}$ system. We have performed DFT simulations modeling the system in a $2 \times 2 \times 2$ unit cell (with one N atom/cell and two $H_2$ molecules/cell, see SI Supplementary Note 5 and Fig. 2), which, although does not account for structural disorder found in MLFF-MD simulations, is still representative to study the effects induced by both N and $H_2$. We optimized a variety of metallic structures including two $H_2$ molecules per unit cell, inspired by the MLFF-MD results or by randomly placing them in the unit cell (the analysis of the disordered

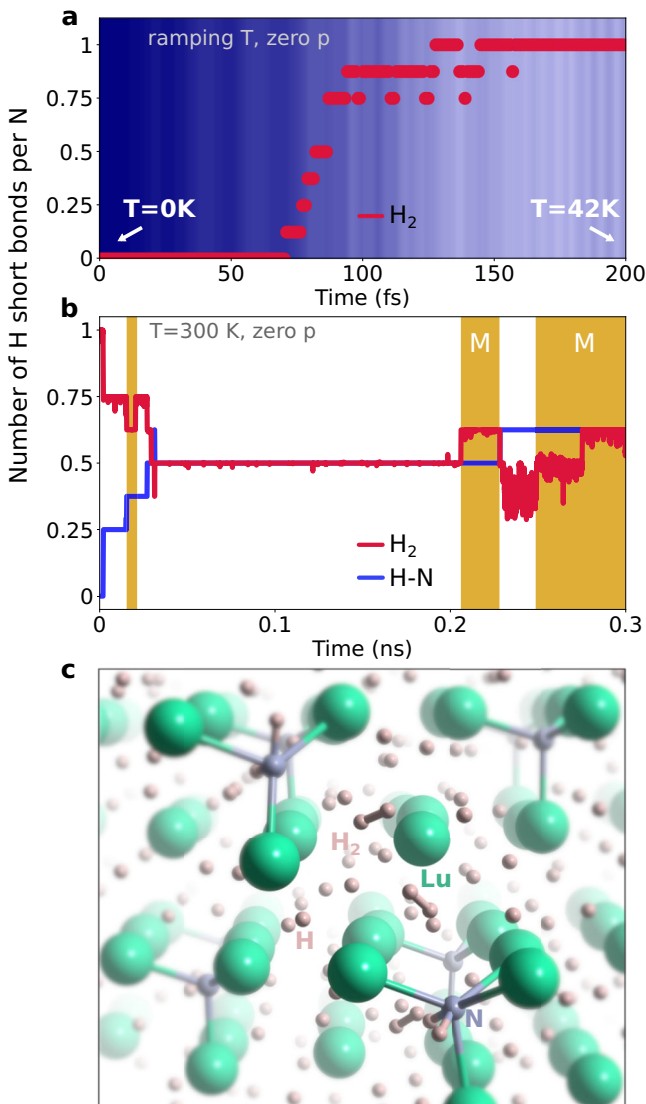

**Fig. 1 | H₂ molecules in the machine-learning-accelerated molecular dynamics (MLFF-MD) simulations. a** Formation of $H_2$ molecules at low temperature; the circles indicate the number of H-H pairs found at every time step below a threshold distance of 1 Å; the effective temperature is indicated as background color gradient (no external pressure $p$ was applied, temperature ranging from 0 to 42 K, see Supplementary Fig. 1 for higher values up to 400 K). **b** MLFF-MD run at T = 300 K and $p$ = 0; the red and blue lines represent the running average (calculated over 100 fs) of the number of $H_2$ molecules (as defined in **a**) and the number of H-N bonds (with a distance < 1.3 Å). The background areas in gold color (with the label 'M' in the larger ones) indicate the metallic regimes. **c** Snapshot of the MLFF-MD showing the disordered phase with H-H and N-H bonds.

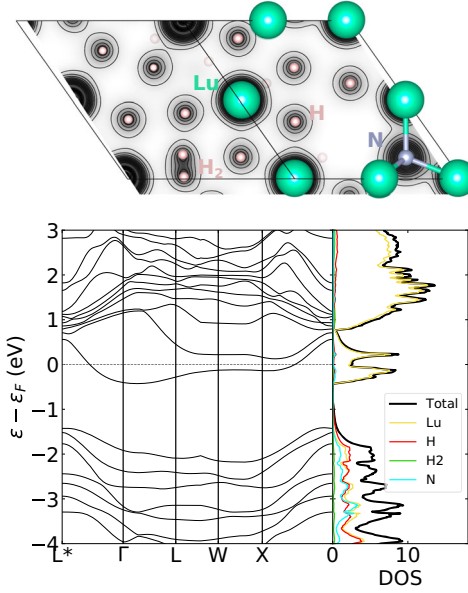

**Fig. 2 | Electronic properties of the representative 2 × 2 × 2 system.** Top: a perspective sketch of the crystal structure of the representative $LuH_{2.875}N_{0.125}$ system in the presence of $H_2$ molecules: charge density on the plane containing one molecule is shown in gray scale (plane belonging to the (10$\bar{1}$) family). Bottom: the relative electronic band structure and projected density of states.

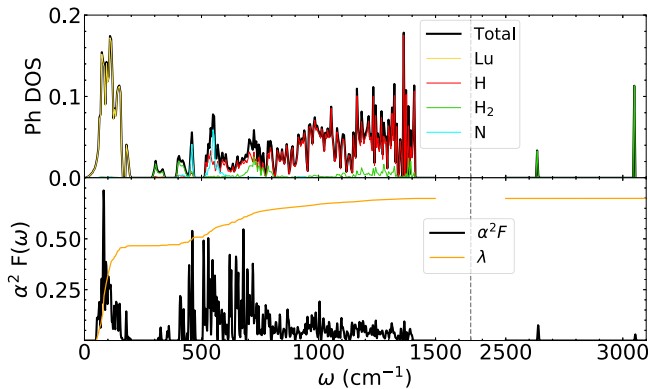

**Fig. 3 | Dynamical and superconducting properties of the representative 2 × 2 × 2 system.** Top: phonon spectrum density of states of $LuH_{2.875}N_{0.125}$ in the presence of $H_2$ molecules. The character of eigenvalues is highlighted (in red the total H character, in green the molecular contribution, in cyan the nitrogen one, and in yellow the total Lu character). Bottom: the evaluated Eliashberg function ($\alpha^2F(\omega)$) and the electron phonon coupling constant ($\lambda(\omega)$ in orange).

structures can be found in the Supplementary Information, Supplementary Note 5).

The electronic density of states, Fig. 2, shows two Lu-derived flat bands and van Hove singularities close to the Fermi level (see also Supplementary Figs. 7 and 9), which may be linked to the emergence of superconductivity. These features are driven by the formation of $H_2$ molecules: In fact, they are not present in the molecule-free LuH-N structures proposed and investigated in the recent literature[20,22,23,28–31].

The dynamical properties of $LuH_{2.875}N_{0.125}$, Fig. 3, confirm the stability of the molecular phase, even at the harmonic level and experimental-ambient pressure (i.e., no imaginary frequencies, see also Supplementary Fig. 14): This is a far from trivial result that underpins the role of molecular hydrogen in the thermodynamic

stabilization of the system, since the molecular-free Fm$\bar{3}$m phase[20,22,47–49] is dynamically unstable[31,38–40].

The phonon frequencies, Fig. 3 (see also Supplementary Fig. 14), are characterized by Lu-derived modes up to ~250 cm⁻¹ and an intermediate frequency range (between 250 and 1500 cm⁻¹) dominated by translational and librational hydrogen modes, while nitrogen contribution is limited to frequencies around 500 cm⁻¹. The high frequency part of the spectrum from ~2800–3000 cm⁻¹ comprises the Raman active vibrational modes of the $H_2$ molecules, strongly renormalized with respect to that of the gas phase[18,56,57]. Interestingly, measured Raman spectra presented in refs. 20,22,49,58,59 shows broad peaks at ~300–800 cm⁻¹ and 3000 cm⁻¹, whose origin were not explicitly addressed and which could be interpreted as translational-like and vibrational-like modes, indicating the presence of $H_2$ molecules in the compounds (see SI for more details). The presence of high-

frequency Raman signals ($\omega \gtrsim 2000$ cm$^{-1}$) can therefore be used as a test for the presence of molecular hydrogen in hydrides.

We can predict the superconducting properties of LuH$_{2.875}$N$_{0.125}$ phase evaluating the Eliashberg spectral function ($\alpha^2 F(\omega)$, Fig. 3) resulting in a total electron-phonon coupling $\lambda = 0.66$, mainly originating from the low-energy Lu-H modes, and from the H$_2$ translational modes (in interesting analogy with what is found in metallic molecular hydrogen[36,46]). The estimation of T$_C$ with the SuperConducting Density Functional Theory[60–62] (see SI for details) gives T$_C \simeq 13$ K, clearly too far from room-temperature but, being obtained for a LuH$_3$-N phase at experimental-ambient pressure, it represents a major result.

In summary, this work proposes a novel paradigm for exploring the physical properties of hydrides at ambient pressure. We have disclosed the role of nitrogen in promoting the formation of H$_2$ units in LuH$_3$. These molecular phases are characterized by a strong disorder and the appearance of different electronic properties strongly linked with the formation of molecular hydrogen, which could determine anomalies in the resistivity measurements: Insulating phases coexist with interesting metallic ones characterized by strongly-coupled low-energy molecular translational modes and low-energy flat electronic bands close to the Fermi level. Finally, the presence of low-energy (degenerate) metastable phases associated with translational and rotational disorder of H$_2$ molecules could bring the system at the verge of structural phase transitions possibly favouring superconducting phases.

We conclude calling for experimental verification of possible presence of hydrogen in molecular form, their dependence on temperature and pressure and their role in determining electrical resistivity. We emphasize the importance of a fine control over the sample preparation, since our study highlights the crucial role played by disorder in determining the electronic properties of hydrides. The possibility to synthesize hydrides at ambient pressure can surely favor the application of experimental techniques impractical at high-pressure superconducting hydrides like Nuclear Magnetic Resonance, muon, neutron and photoemission spectroscopy. The emergence of a low-temperature superconductivity driven by H$_2$ molecules stabilized by N impurities could also stimulate further theoretical studies inspecting the role of pressure, local dis-homogeneity of H, and/or different amount/type of doping with respect to the stability of the molecular phase, seeking for an enhancement of the critical temperature: probably, in the future, artificial intelligence will further aid computational investigations in accounting for the role played by disordered phases[42,63–70].

## Methods

### Machine-learning-accelerated molecular dynamics

The machine-learning-accelerated molecular dynamics (MLFF-MD) simulations were performed by using the Force Field routines[71,72] as implemented in the Vienna Ab Initio Simulation Package VASP[73–75]. We modeled LuH$_{2.875}$N$_{0.125}$ using a $4 \times 4 \times 4$ supercell (with 64 Lu, 184 H, 8 N atoms). We employed the Langevin thermostat[76,77] in the NpT ensemble[78,79], with time steps of 1 fs and zero external pressure.

We first performed thermalization calculations starting from the highly symmetric structure of LuH$_{2.875}$N$_{0.125}$, ramping the temperature from very low temperatures ( <1 K) up to 400 K (50 · 10$^3$ steps). Then, we performed three additional simulations fixing the temperature at 100, 200 and 300 K, separately (300 · 10$^3$ steps per simulation). In all our (ramping and fixed temperature) calculations, we use the on-the-fly training mode as implemented in VASP: Force predictions from the machine-learning force field are used to drive the molecular dynamics simulation; however, if the error estimation at any time step is larger than a threshold value, then a density functional theory (DFT) calculation is performed instead, and the results are used to improve the machine learning force field[71,72]. The threshold to trigger the DFT calculation in the MLFF-MD run is a variable value, automatically determined in VASP: Our convergence tests are discussed in SI (see Supplementary Fig. 18). For the density functional theory component,

we adopted the generalized gradient approximation (GGA) within the Perdew, Burke, and Ernzerhof (PBE) parametrization[80] for the exchange and correlation term, with the $f$ orbitals of Lu atoms excluded from the valence states. We used an energy cutoff of 600 eV, and a $3 \times 3 \times 3$ mesh to sample the Brillouin zone. This setup was employed also in the calculations for the Bader charge (using a finer $6 \times 6 \times 6$ reciprocal-space grid for the smaller $2 \times 2 \times 2$ unit cells, to maintain the same density of sampling points).

We used VESTA[81] for the graphical representation of atomic structures.

### Electronic and phononic properties

Electronic and superconducting calculations were performed using the plane-wave pseudopotential DFT QUANTUM-ESPRESSO package[82–84]. We used ultrasoft pseudopotential[85] for Lu including 5$s$, 6$s$, 5$p$, 6$p$ and 5$d$ states in valence, Optimized Norm-Conserving Vanderbilt pseudopotential[86–88] for hydrogen and nitrogen, and the GGA-PBE approximation, with an energy cut-off of 90 Ry (1080 Ry for integration to the charge).

Integrations over the Brillouin Zone (BZ) of the LuH$_3$ Fm$\bar{3}$m structure were carried out using a uniform $12 \times 12 \times 12$ grid, scaled down for supercells thus ensuring the same sampling density for every system, and a 0.01 Ry Gaussian smearing.

We relaxed Fm3m LuH$_3$ obtaining a lattice parameter of 5.011 Å, in agreement with experimental data[20–22,47–49]. The energy cut-off was enhanced to 120 Ry to ensure the convergence on pressure and the threshold on forces was reduced to 10$^{-5}$ (a.u.). The results shown in the main text have been obtained by adopting a $2 \times 2 \times 2$ supercell using the experimental lattice parameter. Similar results can also be obtained for a fully relaxed supercell including H$_2$ molecules (see Supplementary Fig. 10).

All phonon frequencies and electron-phonon matrix elements were calculated at the harmonic level on the $2 \times 2 \times 2$ supercells, using the linear response theory[82–84], on a $2 \times 2 \times 2$ grid to which correspond 8 $q$-points in the irreducible BZ and a $6 \times 6 \times 6$ mesh for the electronic wavevectors, enhanced to $14 \times 14 \times 14$ mesh for the electron-phonon calculations.

In all calculations (Quantum-Espresso and VASP) we adopted the PBE functional with no additional correction to the electronic correlation: The reliability of the results is discussed in Supplementary Note 6 in the Supplementary Information.

## Data availability

Data supporting the findings of this study are available on https://doi.org/10.6084/m9.figshare.24960708 or from the corresponding authors (C.T. and M.R.) upon request.

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

## Acknowledgements

C.T., P.M.F. and G.P. acknowledge support from CINECA. The computational results presented have been achieved in part using the Vienna Scientific Cluster (VSC). Research at SPIN-CNR has been funded by the European Union - NextGenerationEU under the Italian Ministry of University and Research (MUR) National Innovation Ecosystem grant ECS00000041 - VITALITY, C.T. acknowledges Università degli Studi di Perugia and MUR for support within the project Vitality. C.T. acknowledges the Erwin Schrödinger International Institute for Mathematics and Physics, University of Vienna for the funding received within the "junior research fellowship" and the financial support from the Italian Ministry for Research and Education through PRIN-2022 project "DARk-mattEr-DEVIces-for-Low-energy-detection - DAREDEVIL" (IT-MIUR Grant No. 2022Z4RARB). This research was funded in part by the Austrian Science Fund (FWF) 10.55776/F81 project TACO. For Open Access purposes, the authors have applied a CC BY public copyright license to any author accepted manuscript version arising from this submission. C.F. acknowledges the joint FWF-VDSP "DCAFM DOC 85 doc.funds" project, the FWF TACO (10.55776/F81), and the I4506 FWO-FWF joint project. G.P. acknowledges financial support from the Italian Ministry for Research and Education through PRIN-2017 project "Tuning and understanding Quantum phases in 2D materials - Quantum 2D" (IT-MIUR Grant No. 2017Z8TS5B) and fundings from the European Union - NextGenerationEU under the Italian Ministry of University and Research (MUR) National Innovation Ecosystem grant ECS00000041 - VITALITY - CUP E13C22001060006. L.R. was supported by the Austrian Science Fund (FWF), projects I 4506 (FWO-FWF joint project). We thank A. Sanna for useful discussions.

## Author contributions

C.T., G.P. and P.M.F. conceived the idea. C.T. and M.R. supervised the project. A.A., L.R. and M.R. performed the MLFF-MD calculations. P.M.F. and C.T. performed the DFT/DFPT calculations. All authors contributed to the final version of the manuscript, and to the discussion and interpretation of the results.

## Competing interests

The authors declare no competing interests.
