## [Peer Review File · Nature Communications]

Molecular Hydrogen in the N-doped LuH₃ System as a Possible Path to SuperconductivityREVIEWER COMMENTS

Reviewer #1 (Remarks to the Author):

Cesare Tresca et al. investigated the Evidence of Molecular Hydrogen in the N-doped LuH₃ System and proposed a Possible Path to Superconductivity. Using machine learning-accelerated force-field molecular dynamics, the authors investigated the formation of H₂ molecules in nitrogen-doped lutetium hydride, demonstrating the active role of nitrogen in stabilizing this phase. Further, the DFT calculations showed a dynamically stable system and superconductivity below 10 K at ambient (or no) pressure. The authors proposed a possible route to stabilize the molecular hydrogen in LuH₃ at near-ambient conditions. Overall, this is an interesting computational study but lacks scientific discussion. The present form of the manuscript is not suitable for nature communication. I have a few queries which may improve the clarity of the manuscript:

- 1: The authors have discovered the new structure where the H₂ molecule is formed. What is the space group of the structure of this new (metastable) phase?
- 2: NpT simulation on a smaller cell leads to large volume fluctuations; how have the authors managed to manage this significant volume fluctuation?
- 3: Why has the author used a smaller unit-cell for phonon and electron-phonon coupling calculation?
- 4: The authors stated, "The system exhibits a high degree of disorder, as also expected in the real samples, with the molecules randomly distributed." how does a single configuration, particularly on a smaller unit cell, represent the highly disorder system (H₂ orientation). I want authors to validate their results on different disorder configurations to ensure the robustness of their claim.
- 5: The authors claim that the low-energy liberational modes show strong electron-phonon coupling, contributing most to superconductivity. What causes these low-energy modes to be strongly coupled with flat-electronic bands? Also, the low-energy modes and the flat electronic bands are strongly contributed by Lu and not by hydrogen; I am wondering what is the role of H₂ molecules in T_c?
- 6: In the main manuscript, the authors mentioned that the N-doped system is dynamically stable in the metastable phase. The author should compare the energy of the two phases

(i.e. LuH_{2.875}N_{0.125} with one H₂ molecule and LuH_{2.875}N_{0.125} in its metastable phase) and comment on their static and dynamical stability.

7: Finally, the authors need to comment on the possible strategy to enhance the T_c based on the present study.

Reviewer #2 (Remarks to the Author):

The authors present an extremely timely and interesting study about the path to superconductivity in the controversial N-doped LuH₃ system, where they propose molecular Hydrogen as a potential mechanism for superconductivity. This paper is extremely thorough and well written and provides new insight to a very hot topic in the condensed matter field. Although I believe this manuscript is publishable, certain major revisions can be made to strengthen the paper for a prestigious journal such as Nature Communications.

1. The authors mention artificial intelligence and data-driven approaches as a future pathway to identify new candidate superconductors (line 71) but only cite one review paper from 2021 (ref. 30). There has been a plethora of works in the past few years using these methodologies. It would benefit the paper if the authors could add these citations to the introduction and briefly discuss.

1. <https://arxiv.org/abs/2212.06071>

2. <https://www.sciencedirect.com/science/article/pii/S092702562300352X>

3. <https://www.nature.com/articles/s41524-022-00933-1>

4. <https://pubs.acs.org/doi/full/10.1021/acs.nanolett.2c04420>

5. <https://arxiv.org/abs/2307.10728>

2. The authors use a substitutional N doping concentration of 12.5 % in the LuH₃ structure, can you comment on how realistic or how achievable this doping concentration might be in

experiments? How does this doping concentration compare to the recent and controversial experimental synthesis?

3. I was interested by the two flat bands and van Hove singularities close to the Fermi level. How can you definitively say that these are promoted by the formation of H₂ molecules? Is it possible that these flat bands are due to some sort of strong correlation? Can the authors comment more on this topic and maybe add some relevant references.

4. I find it very interesting how different the electronic structure is of the two proposed 2x2x2 representative systems are. You discuss dynamical stability, but it would be useful to calculate formation energy and energy above the convex hull of these compounds and compare to other Lu-H-N structures in literature or DFT databases. This will aid in the discussion of stable/metastable structures. In addition, can the authors comment on a strategy to further explore potential Lu-H-N structures with various H₂ and N doping concentrations (i.e. enumerating supercells, varying concentrations, possibly using cheaper machine learning-based tools to screen candidate structures)? The authors only focus on these two possibilities, but I am sure there are more that can be explored in future work. Providing some insight on how to move forward down this path would be extremely useful.

5. The authors report a relatively low transition temperature of the metallic 2x2x2 Lu-H-N system (~10 K) at ambient conditions. Although this temperature is low, it is still an extremely novel result. It has been reported that applying even small amounts of pressure can increase the transition temperature of hydride materials substantially. If it is feasible, would it be possible for the authors to apply various amounts of pressure at the DFT level and compute the T_c? This could be an extremely novel result if you find that small amounts of pressure (<100 Gpa) drastically increases the T_c. In addition, this could further aid in establishing the roadmap to explore other candidate superconductors in the future.

6. The authors state that the data that supports the findings can be available upon reasonable request. Building on the theme of transparency and reproducibility (that has been missing from previous LuH₃ works), I request that the authors make all the data available in public repositories such as Figshare or GitHub.

Reviewer #3 (Remarks to the Author):

The manuscript employs machine-learning aided molecular dynamics to investigate the metastable phase of the Lu-N-H system. It delves into potential H₂ dimerization mechanisms and examines the electron-phonon interaction properties of a representative structure. Nevertheless, the study does not conclusively establish the enhancement of T_c by H₂ units. As such, it may not meet the publication criteria for a journal at the level of Nature Communications.

Specific Comments:

1. The slight excess of charges on the H₂ molecule leading to longer H-H bond lengths may be attributed to a self-interaction error phenomenon. This error arises from the over-delocalization of electrons on the Lu (Lutetium) atoms, causing them to populate antibonding orbitals. This issue holds significance in various Lu-N-H systems, as evident in research articles such as arXiv preprints arXiv:2306.11511 (2023), arXiv:2306.09868 (2023), and arXiv:2305.18196 (2023). In the current manuscript, it is noted that the valence state of Lu undergoes frequent changes, highlighting the necessity for a rigorous treatment of electron correlation in the analysis.

2. There is a lack of compelling evidence to support the notion that the formation of H₂ molecules enhances the superconducting transition temperature (T_c). Figure 2 does not indicate any discernible impact of H₂ on the electronic structure around the Fermi level, while Figure 3 suggests that the H₂ peaks make no substantial contribution to the overall electron-phonon coupling. Consequently, the T_c observed does not exceed that of numerous Lu-N-H systems lacking H₂ units, indicating that the presence of H₂ does not lead to a notable increase in T_c.

Responses to Referees' comments for the paper
***"Evidence of Molecular Hydrogen in the N-doped LuH₃ System:
a Possible Path to Superconductivity?"***

Cesare Tresca,^{1,*} Pietro Maria Forcella,² Andrea Angeletti,^{3,4} Luigi
Ranalli,^{3,4} Cesare Franchini,^{4,5} Michele Reticcioli,^{4,†} and Gianni Profeta^{2,1}

¹*CNR-SPIN c/o Dipartimento di Scienze Fisiche e Chimiche,
Università degli Studi dell'Aquila, Via Vetoio 10, I-67100 L'Aquila, Italy*

²*Dipartimento di Scienze Fisiche e Chimiche,
Università degli Studi dell'Aquila, Via Vetoio 10, I-67100 L'Aquila, Italy*

³*University of Vienna, Vienna Doctoral School in Physics,
Boltzmannngasse 5, 1090 Vienna, Austria*

⁴*University of Vienna, Faculty of Physics and Center for Computational Materials Science,
Kolingasse 14-16, 1090 Vienna, Austria*

⁵*Dipartimento di Fisica e Astronomia, Università di Bologna,
Viale Berti Pichat 6/2, I-40127 Bologna, Italy.*

* cesare.tresca@spin.cnr.it

† michele.reticcioli@univie.ac.at

Point-by-point replies to Referees' Reports

(In the new version of the paper, changes are marked in red.)

Reviewer: A

Comment:

Cesare Tresca et al. investigated the Evidence of Molecular Hydrogen in the N-doped LuH₃ System and proposed a Possible Path to Superconductivity. Using machine learning-accelerated force-field molecular dynamics, the authors investigated the formation of H₂ molecules in nitrogen-doped lutetium hydride, demonstrating the active role of nitrogen in stabilizing this phase. Further, the DFT calculations showed a dynamically stable system and superconductivity below 10 K at ambient (or no) pressure. The authors proposed a possible route to stabilize the molecular hydrogen in LuH₃ at near-ambient conditions. Overall, this is an interesting computational study but lacks scientific discussion. The present form of the manuscript is not suitable for nature communication. I have a few queries which may improve the clarity of the manuscript:

Answer:

We thank the Referee for careful reading our manuscript, and for highlighting all the original results, namely: 1) Predicting formation of H₂ molecules by means of machine learning-accelerated force-field molecular dynamics. 2) Clarification of the role of nitrogen in Lu-N-H system. 3) Discovery of a possible metastable superconducting molecular phase showing a superconducting phase.

We have followed the Referee's suggestions to improve the clarity of the manuscript, as discussed below.

Point: 1

The authors have discovered the new structure where the H₂ molecule is formed. What is the space group of the structure of this new (metastable) phase?

Answer:

We take the opportunity to make some computational point clearer, especially for different scientific communities. As written in the main text, we described the system considering possible disorder effects in the structure. This aspect is indeed not always considered in other structural search studies on the same system.

Thus, the structures containing H₂ molecules are intrinsically disordered, without any structural symmetries, thus the space group is P1. We have explicitly added this information in the Supplementary Information document. Moreover, we decided to make all our structures available as supplementary material.

We would like to further underline that treatment of disorder is crucial in describing the Lu-N-H_x system; the structural information obtained by XRD experiments (see [Nature 615, 244 (2023), Nature 620, 72 (2023), AIP Advances 13, 065117 (2023), Phys. Rev. B 108, 214505 (2023)]) confirm that the metal subsystem (Lu-N matrix) retains the Fm $\bar{3}$ m symmetry, because hydrogen atoms are not revealed by XRD experiments. To date, the only experimental technique revealing the presence of hydrogens is Raman spectroscopy (see [Nature 615, 244 (2023), Nature 620, 72 (2023), Phys. Rev. B 108, 214505 (2023)]), resulting in a signal in the 500-700 cm⁻¹ and 3000-3600 cm⁻¹ regions, which can be interpreted as to the presence of hydrogen in molecular form, as suggested in our work. We have included in the main text the citations mentioned above.

Point: 2

NpT simulation on a smaller cell leads to large volume fluctuations; how have the authors managed to manage this significant volume fluctuation?

Answer:

This is a very good point that was indeed already examined during the molecular dynamics simulations, although not explicitly mentioned in the original manuscript. Once the temperature is stabilized at the desired value, the volume oscillates only 2% around the average volume value throughout the entire NpT simulation. This stability was achieved by adopting a standard procedure, *i.e.*, using the Langevin thermostat with a fictitious mass for the lattice degrees-of-freedom of 1000 amu and a friction coefficient for lattice degrees-of-freedom of 10 ps^{-1} . We have now included these information and the corresponding plot in the Supplementary Information document (figure SF3). The plot shows the volume variation in MLFF-MD time windows corresponding to the transition between the insulating and metallic phase, and it is representative of the whole run. Moreover, we have uploaded as supplementary material all structures from the whole MLFF-MD run at 300 K.

Point: 3

Why has the author used a smaller unit-cell for phonon and electron-phonon coupling calculation?

Answer:

We thank the Referee for this comment, giving us again the possibility to improve the description of the computational details. The $\text{LuH}_{2.875}\text{N}_{0.125}$ unit cell we adopted in machine-learning-accelerated molecular dynamics simulations contains 64 Lu, 184 H and 8 N atoms for a total of ~ 250 atoms, which allowed us to model the high degree of disorder due to N impurities and molecular hydrogen. Although the size of the system is still treatable with standard DFT calculations, it represents a formidable task for the phonon and electron-phonon calculations required to analyze the superconductive properties, as it is extremely demanding from a computational point of view (in fact, we are not aware of any electron-phonon first-principles calculation with comparable setup).

Therefore, adopting a smaller unit cell in order to investigate the superconducting properties of the newly discovered structures was a necessary choice, dictated by computational limitations. We think that the chosen unit cell represents a reasonable trade off between computational feasibility and accuracy in the description of the superconducting properties. Indeed, tests calculations considering other configurations of hydrogens and molecules, arranged differently in space, (which however show higher or similar formation energies with respect to the one considered in the main paper) results in essentially the same physical properties.

We clarified this aspect in the Supplementary Materials, Sec. V.

Point: 4

The authors stated, “The system exhibits a high degree of disorder, as also expected in the real samples, with the molecules randomly distributed.” how does a single configuration, particularly on a smaller unit cell, represent the highly disorder system (H_2 orientation). I want authors to validate their results on different disorder configurations to ensure the robustness of their claim.

Answer:

We thank the Referee for the legitimate and relevant question. Aware of the possible effects of disorder on our results, we considered different unit cells modeling many H_2 configurations. We identified the lowest energy configurations among all cases inspected, and reported our findings for such structures, which better represent the thermodynamic limit. Fig.1 shows the electronic structures for different configurations of one and two molecules.

FIG. 1. Electronic band structures for alternative $LuH_{2.875}N_{0.125}$ system with one H_2 molecule (left) and two H_2 molecules (right).

We found that all considered molecular phases showed qualitatively the same physical properties of the lowest energy phase, namely the presence of flat bands and stable phonons (for two H_2 alternative systems, see Fig.2). In addition, all the considered (metallic) phases have been found superconducting (with T_c ranging from $\simeq 10$ K to about 20 K). Despite some structures exhibited small phonon instabilities (about -100 cm^{-1} , at specific q-points

in the Brillouin zone) signaling long range distortions of hydrogen atoms, we found only relatively small quantitative differences with the results reported in the manuscript. In order to make this point clearer, we added a specific comment in the SM.

FIG. 2. Phonon dispersion and electron-phonon coupling for the alternative $\text{LuH}_{2.875}\text{N}_{0.125}$ system with two H_2 molecules.

Point: 5

The authors claim that the low-energy librational modes show strong electron-phonon coupling, contributing most to superconductivity. What causes these low-energy modes to be strongly coupled with flat-electronic bands? Also, the low-energy modes and the flat electronic bands are strongly contributed by Lu and not by hydrogen; I am wondering what is the role of H₂ molecules in Tc?

Answer:

We thank the Referee for the appropriate question. We are very sorry to admit that while reading the Referee's comment, we discovered a typo in the manuscript when we refer to the low-energy molecular modes. We wrongly refer them as librational, but they are indeed "translational modes", in which the molecules rigidly move without any appreciable rotational component. However, this typo does not alter at all the discussion and conclusions of the paper.

As far as the answer to the comment: we apologize for the lack of clarity in describing this point in the original version of the manuscript.

As clear from Fig. 3 of the manuscript, the function $\lambda(\omega) = 2 \int_0^\omega \alpha^2 F(\omega') / \omega' d\omega'$ shows two main steps: one at the low-energy Lu-derived modes ($0 \text{ cm}^{-1} \leq \omega \leq 250 \text{ cm}^{-1}$) and the other starting from $\simeq 500 \text{ cm}^{-1}$ and ending at $\simeq 800 \text{ cm}^{-1}$ due to molecular and Nitrogen phonon modes (as evident from comparison with the projected phonon density of states).

These last modes contribute more than 25% to the total λ , thus contributing to the emergence of the (low temperature) superconducting phase.

In order to make this point clearer and motivated by the Referee's comment, we decided to highlight the coupling of different phonon modes with the flat band crossing the Fermi level, highlighting the role of H₂ modes. We have calculated the deformation potential for representative phonon modes (at $\simeq 450 \text{ cm}^{-1}$, $\simeq 550 \text{ cm}^{-1}$ and $\simeq 750 \text{ cm}^{-1}$) in the region where $\lambda(\omega)$ is enhanced (see Fig.3 of the manuscript), highlighting the deformation of the bands at the Fermi energy.

The results are presented in Fig.3(d),(e),(f).

The first mode, represented in Fig.3(a) mainly involve N, H₂ and H displacement and strongly couples with the bands crossing the Fermi level, in particular for the states

FIG. 3. The top row illustrates crystal structures along with eigenvectors corresponding to mode 27 at $\simeq 450 \text{ cm}^{-1}$ (a), 29 at $\simeq 550 \text{ cm}^{-1}$ (b) and 40 at $\simeq 750 \text{ cm}^{-1}$ (c). To enhance clarity, only the eigenvectors making significant contributions are depicted in these figures. In the bottom row are shown the deformations of the bands at the Fermi energy for mode 27 (d), mode 29 (e) and mode 40 (f). In black is represented the pristine band structure for $\text{LuH}_{0.875}\text{N}_{0.125}$ while the color gradient show the deformation moving along the eigenvectors with different percentage on their modulus.

around the Gamma point. The second mode we analyze in Fig. 3(b) does not have components on the H_2 molecule, and indeed shows the lowest coupling (among the three analyzed modes) with the states at the Fermi energy. On the other hand, the last mode in Fig. 3(c) moves only molecular and atomic hydrogen (without N) showing the largest coupling, strongly deforming the bands at the Fermi energy. This analysis highlights how molecules strongly contribute to increase the electron-phonon coupling in the region $\simeq 450 - 750 \text{ cm}^{-1}$ due to the strong deformation potential at the Fermi energy, with consequent impact on the superconducting phase.

Moreover, we note that molecular hydrogen plays a key role in the dynamical stability of the system, as discussed in the reply to Point 6. We apologize for this oversight, which we have corrected in the new version of the manuscript, adding also a short paragraph in the SM section VII.

Point: 6

In the main manuscript, the authors mentioned that the N-doped system is dynamically stable in the metastable phase. The author should compare the energy of the two phases (i.e. LuH₂.875N_{0.125} with one H₂ molecule and LuH₂.875N_{0.125} in its metastable phase) and comment on their static and dynamical stability.

Answer:

We agree with the Referee about the importance of comparing the stability of these two phases. We note that this comparison is already present in the original version of the manuscript, in particular in Section V and VI of the SM.

In the description contained in these two sections, we include the comparison between the formation energies of the phases containing one and two molecules, as well as their electronic and dynamical properties, and discuss them. We have decided to not include these information in the main text to avoid inclusion of details not strictly necessary to understand the manuscript, thus referring to dedicated SM sections in which we could deeply and lengthy discuss the detail of certain aspects which result indeed important and interesting for a specialized audience. However, in the revised version of the manuscript, we have added a comment to reference the discussion in the SM.

Point: 7

Finally, the authors need to comment on the possible strategy to enhance the T_c based on the present study.

Answer:

The Referee raises an interesting point: we have extend the Conclusions in the main text to include a comment on the critical temperature.

We would like to point out that our study enters a controversial discussion about near-ambient superconductivity in Lu-H-N, see the recent retraction of the publication first reporting the high T_c , Nature 615, 244 (2023). All the numerous theoretical studies on the topic have concluded that the system is not superconductive unless a large pressure is applied (in $Fm\bar{3}m$ structure). Conversely, our findings propose a novel concept for realizing the superconductive phase, arising from a disordered phase exhibiting molecular hydrogen. Our results show a critical temperature of approximately 10 K, found in the molecular metastable phase at zero pressure. It is beyond the scope of our work to confirm/support the (retracted) experimental results of near-ambient T_c reported in Nature 615, 244 (2023). Rather, our proof-of-concept study establishes the possibility to induce zero-pressure (low-temperature) superconductivity in Lu-H-N, by modeling the disordered metastable phase as induced by molecular hydrogen.

However, mechanisms to increase T_c can be certainly envisioned. First of all, it would be interesting to explore the effects of external pressure on the disordered Lu-H-N phase: pressure could indeed induce formation of additional H_2 molecule, with potential impact on the superconducting properties.

Moreover, our proof-of-concept work is only the first step in the theoretical study of the disordered phase. The modeling of this phase could be extended in order to include additional effects that might play a pivotal role. For example, larger cells could be employed in the molecular dynamics simulations in order to model larger degree of disorder. Additionally, in this setup, the effects of the disorder of N impurity could also be addressed. Finally, since hydride compounds are known to exhibit dis-homogeneous distributions of H atoms, first-principles simulations could model the local deviation of the H content, and analyze the effect on the stability of the H_2 molecules and

superconductivity.

Reviewer: B

Comment:

The authors present an extremely timely and interesting study about the path to superconductivity in the controversial N-doped LuH₃ system, where they propose molecular Hydrogen as a potential mechanism for superconductivity. This paper is extremely thorough and well written and provides new insight to a very hot topic in the condensed matter field. Although I believe this manuscript is publishable, certain major revisions can be made to strengthen the paper for a prestigious journal such as Nature Communications.

Answer:

We express our gratitude to the Referee for her/his valuable advice, and we are pleased to address her/his requests and criticisms in the following responses.

Point: 1

The authors mention artificial intelligence and data-driven approaches as a future pathway to identify new candidate superconductors (line 71) but only cite one review paper from 2021 (ref. 30). There has been a plethora of works in the past few years using these methodologies. It would benefit the paper if the authors could add these citations to the introduction and briefly discuss.

1. <https://arxiv.org/abs/2212.06071>
2. <https://www.sciencedirect.com/science/article/pii/S092702562300352X>
3. <https://www.nature.com/articles/s41524-022-00933-1>
4. <https://pubs.acs.org/doi/full/10.1021/acs.nanolett.2c04420>
5. <https://arxiv.org/abs/2307.10728>

Answer:

The Referee asks us to consider other references regarding the use of artificial intelligence and data-driven approaches to identify new potentially interesting superconductors. We carefully checked all the cited references, but we think that they do not really fit the way we used the machine learning approach in our work.

In particular, the references listed by the Referee all refer to the use of data-driven approaches to propose possible new superconducting materials by learning critical temperatures from large databases of known superconductors (experimentally verified and/or predicted by first-principles methods), finding proper descriptors to correlate them with different critical temperatures.

On the contrary, in our work we did not use such techniques: In our study, we adopted the machine-learning force field, where the machine learning model acts as an accelerator in the ab-initio molecular dynamics simulations, allowing us to extend the exploration time in our simulations. The computational methodology is completely general and encompasses the application on superconducting materials. Our work, indeed, demonstrated how such advanced computational techniques could indeed be very useful also in the field of superconducting hydrides, where experimental information on the structural phases are difficult to obtain because of the presence of hydrogen, which result "invisible" to X-rays. The reference we included is a "perspective paper" on the challenges we need to solve to

search for higher temperature/lower pressure superconducting hydrides, in which many authors point to the use of machine-learned force fields for molecular dynamics simulations to find proper superconducting phases. We have now added to the introduction references specific to the machine-learning force field methods used in our work, previously cited only in the Methods Section:

doi:10.1103/PhysRevB.100.014105 and

doi: 10.1103/PhysRevLett.122.225701.

Anyway, we added a comment on the possibility to design new high-temperature superconductors through deep-learning approaches in the Conclusions, adding, among others, also the references suggested by the Referee .

Point: 2

The authors use a substitutional N doping concentration of 12.5 % in the LuH₃ structure, can you comment on how realistic or how achievable this doping concentration might be in experiments? How does this doping concentration compare to the recent and controversial experimental synthesis?

Answer:

We have considered the nominal concentration stated and proposed in Dasenbrock-Gammon and collaborator's original work [Nature 615, 244 (2023)], in which they refer to an N-doping concentration of 12.5% in the LuH₃. This concentration was also confirmed by the work of Salke et al. [arXiv:2306.06301-1], who performed additional measurements on Dasenbrock-Gammon's samples.

In order to clarify this aspect, we have amended the manuscript: "a substitutional N doping on H sites of 12.5% (in line with the content reported for the experimental samples) [Dasenbrock-Gammon et al., Salke et al.]".

Point: 3

I was interested by the two flat bands and van Hove singularities close to the Fermi level. How can you definitively say that these are promoted by the formation of H₂ molecules? Is it possible that these flat bands are due to some sort of strong correlation? Can the authors comment more on this topic and maybe add some relevant references.

Answer:

We thank the Referee for underlining this aspect, giving us the possibility to further comment on this point. The appearance of flat-bands close to the Fermi level is strongly linked to the formation of hydrogen molecules, promoting peculiar doping effects interacting with Nitrogen dopants (as explained in the manuscript, van Hove singularities are not present in the pristine compound with no molecules, as reported in several publications, *e.g.*, Nat. Comm. 14, 5367 (2023)). Thus, their origin is chemical, related to peculiar structural/electronic properties of the considered phases. On the contrary, their appearance cannot originate from "strong-correlations" effects (we interpret this as correlation effects beyond those considered in the exchange-correlation functional we adopted), as correction to the correlations were not included in our calculations. Indeed, to the best of our knowledge, nearly all the computational predictions on superconducting hydrides (at low and high pressure) were conducted without invoking (or adding) additional correlation effects apart those already considered in the local density (or gradient corrected) functional. The prediction of different new superconducting materials that were later verified by the experiments certifies the validity of the theoretical and computational approach. Moreover, as also requested by Referee C in Point 1, we analyzed the effects of including corrections to the electronic correlation in the computational setup, using the DFT+U approach: By comparing the DFT+U and original DFT data, we observe no qualitative difference in the results. This analysis is now included in the Supplementary Information document: Section VI for the discussion of the effects of the U parameters, and the analysis of the electronic properties of the molecular hydrogen.

Point: 4

I find it very interesting how different the electronic structure is of the two proposed $2 \times 2 \times 2$ representative systems are. You discuss dynamical stability, but it would be useful to calculate formation energy and energy above the convex hull of these compounds and compare to other Lu-H-N structures in literature or DFT databases. This will aid in the discussion of stable/metastable structures. In addition, can the authors comment on a strategy to further explore potential Lu-H-N structures with various H₂ and N doping concentrations (i.e. enumerating supercells, varying concentrations, possibly using cheaper machine learning-based tools to screen candidate structures)? The authors only focus on these two possibilities, but I am sure there are more that can be explored in future work. Providing some insight on how to move forward down this path would be extremely useful.

Answer:

We thank the Referee for this question. The electronic structures shown in the main text, as well as in Fig. SF7 in the Supplementary Materials, refer to Lu-H-N compounds with identical stoichiometry: In all these cases, the structures exhibit 8, 23, and 1 atoms of Lu, H and N, respectively (corresponding to a N-doping concentration of 12.5%, as reported in the manuscript). The difference between the various cases regards the number of molecules formed by H atoms in the compound: Note that this molecular hydrogen does not originate from extrinsic doping, but from H atoms originally included in the crystal. The inspection of the stability of different stoichiometry via the convex-hull is surely interesting, but beyond the scope of this work: Our proof-of-concept study predicts the stabilization of a (low-temperature, zero-pressure) superconductive phase upon formation of molecular hydrogen in the compound. Convex-hull investigations could possibly lead to Lu-H-N structures (with different stoichiometry) showing higher superconductive critical temperature. Convex-hull studies have been already performed on this compound, without obtaining any zero-pressure superconductivity: These investigations were indeed carried out without focusing on the possibility for hydrogen to form H₂ complexes.

We believe that by carefully including this possibility, convex-hull investigations could lead to phases with enhanced critical temperature, as compared to the one reported in our study. We have included a comment in the Conclusions of the revised manuscript in order to suggest this possible exploration route, as suggested by the Referee.

Point: 5

The authors report a relatively low transition temperature of the metallic 2x2x2 Lu-H-N system (~ 10 K) at ambient conditions. Although this temperature is low, it is still an extremely novel result. It has been reported that applying even small amounts of pressure can increase the transition temperature of hydride materials substantially. If it is feasible, would it be possible for the authors to apply various amounts of pressure at the DFT level and compute the T_c ? This could be an extremely novel result if you find that small amounts of pressure (<100 Gpa) drastically increases the T_c . In addition, this could further aid in establishing the roadmap to explore other candidate superconductors in the future.

Answer:

We thank the Referee for the positive evaluation of our work, and for suggesting a possible strategy for the enhancement of the critical temperature. The application of pressure might indeed turn beneficial to this aim. However, the scope of this study is to report the onset of a superconductive phase at no pressure, driven by N-induced molecular hydrogen, a result that offers new and interesting perspectives for new studies and approaches on these fast developing subjects.

Therefore, we prefer to avoid to include high-pressure data in this publication, but we have added a comment in the Conclusions, in order to underline further potential research routes.

Point: 6

The authors state that the data that supports the findings can be available upon reasonable request. Building on the theme of transparency and reproducibility (that has been missing from previous LuH3 works), I request that the authors make all the data available in public repositories such as Figshare or GitHub.

Answer:

We gladly accept the Referee's recommendation to transparently share the data. We have uploaded the most relevant information as Supplementary Datasets: files including structural information of the systems analyzed in our study (for both phonon/superconducting calculations and molecular dynamics), the main output files of the MLFF-MD production run was uploaded in figshare, due to the file size. Any further data remains available upon reasonable requests to the authors, as per Nature Communications guidelines.

Reviewer: C

Comment:

The manuscript employs machine-learning aided molecular dynamics to investigate the metastable phase of the Lu-N-H system. It delves into potential H₂ dimerization mechanisms and examines the electron-phonon interaction properties of a representative structure. Nevertheless, the study does not conclusively establish the enhancement of T_c by H₂ units. As such, it may not meet the publication criteria for a journal at the level of Nature Communications.

Answer:

We thank the Referee for her/his work in reviewing the paper, which is an incentive for us to go deeper in our results. However, we respectfully disagree with her/his opinion. In fact, her/his statement "the study does not conclusively establish the enhancement of T_c by H₂ units" is not correct.

First, we have demonstrated the formation of H₂ molecules in the Lu-N-H compound. Then, we highlighted the crucial role of molecular hydrogen in the dynamical stabilization of the compound within the Fm-3m unit cell for the Lu sublattice. These results provide the community with an unprecedented explanation of the characterization measurements (i.e., XRD and Raman measurements) reported in [Nature 615, 244 (2023) , Nature 620, 72 (2023), AIP Advances 13, 065117 (2023), Phys. Rev. B 108, 214505 (2023)]. Finally, at variance with all the great variety of theoretical studies published in the recent months, we reported the emergence of a zero-pressure superconductive (metastable) phase: The compounds exhibits such superconductivity only in the molecular hydrogen phase stabilized by the N impurities, as explained in the main text.

Moreover, we found that molecular hydrogen "translational modes" strongly entangled with N-related modes contribute by ~25% to the electron-phonon coupling (λ), which account for T_c \simeq 10 K (as discussed in the main text). The role of H₂ in the emergence of the superconductive phase is, thus, clear: detailed information about this point can be found in the SI, Section VII (also as part of the the answer to Referee A, point 5).

As mentioned also by the other Referees, although the critical temperature of the

molecular-hydrogen phase that we found is low, the zero-pressure and the N-H₂-induced mechanism reveal novel Physics which will open a route in the exploration of high temperature superconductivity in hydrides. Our contribution adds new and important ingredients that might be exploited for the comprehension of the physical mechanism of hydrides at ambient conditions.

We take the opportunity to remark that our goal is not to validate the controversial near-ambient-condition measurements on the superconductivity in Lu-H-N [Nature 615, 244 (2023)]. On the contrary, our proof-of-concept study establishes the possibility to stabilize hydrogen in a molecular (metastable) phase via N doping, which in turn leads to the emergence of (zero-pressure, low-temperature) superconductivity. Anyhow, possible strategies to enhance the critical temperatures can be envisioned (e.g., external pressure, varying the N-vs-H content), and are discussed in the revision version of the manuscript (see Conclusions).

Point: 1

The slight excess of charges on the H₂ molecule leading to longer H-H bond lengths may be attributed to a self-interaction error phenomenon. This error arises from the over-delocalization of electrons on the Lu (Lutetium) atoms, causing them to populate antibonding orbitals. This issue holds significance in various Lu-N-H systems, as evident in research articles such as arXiv preprints arXiv:2306.11511 (2023), arXiv:2306.09868 (2023), and arXiv:2305.18196 (2023). In the current manuscript, it is noted that the valence state of Lu undergoes frequent changes, highlighting the necessity for a rigorous treatment of electron correlation in the analysis.

Answer:

We thank the Referee for this interesting comment. In the works cited by the Referee, researchers included an Hubbard repulsion on f and d orbitals of Lu to mimic the results obtained using hybrid functional for LuX compounds in a (cheaper) DFT+U approach [arXiv:2306.11511 (2023); arXiv:2306.09868 (2023); arXiv:2305.18196 (2023)].

In particular, the Hubbard-like on-site repulsion on the Lu f orbitals for Lu hydrides is taken into account in all the cited references, while the contribution on the d orbitals appears in one of the three proposed works [arXiv:2305.18196 (2023)].

Let us analyze the first contribution on the Lu-4 f orbitals. As also found in previous works [arXiv:1408.0863 (2014), arXiv:2306.11511 (2023)] an effective U of about 5 ÷ 7 eV (depending on the phase considered) is needed to properly predict the energy position of the f -orbital, which results deep in energy ($\sim -6 \div -8$ eV below the Fermi level), thus not active close to the Fermi energy. Due to their deep energy, these states do not influence our analysis: Thus, we can safely exclude f -states from the valence states in the pseudopotential (as routinely done by the majority of the first-principles calculation on lanthanides). Therefore, no Hubbard correction is required, since the f states are simply excluded. To confirm the validity of this approach, we performed additional calculations, following the Referee's suggestion: we report in Fig. R4 the density of states of our representative unit cell (see Fig.2 of the manuscript), with (and without) the f -states in valence corrected with U= 5.5 eV (in line with Nature 615, 244 (2023) and arXiv:1408.0863 (2014)) compared with the same calculation but without the Hubbard term. As evident,

FIG. 4. Comparison between the densities of state for the $\text{LuH}_{2.875}\text{N}_{0.125}$ system with two H_2 molecules with and without the Hubbard correction on f states ($U_f = 5.5$ eV) in black and red respectively. For comparison we report also the DOS obtained without the inclusion of f -states in valence (cyan).

the effect of f states is limited mainly in the region of $\simeq -7$ eV, while the region around the Fermi level (relevant for our main results) is qualitatively unaffected and essentially indistinguishable by the electronic properties obtained without f -states in valence.

We now move to consider the role of the d states. Also in this case, we performed additional calculations. The Hubbard term proposed for Lu- $5d$ orbitals [arXiv:2305.18196 (2023), arXiv:1408.0863 (2014)] is 8.2 eV for Lu d -states fitting this value on calculation using hybrid functional on LuN system, without hydrogens. Given that the Hubbard term suggested for Lu- $5d$ orbitals is determined through a fitting procedure on hybrid functionals in a system substantially different from ours, we have opted to rigorously re-calculate the Hubbard term value in our system using first principles calculations: To this aim, we employed the method proposed by Matteo Cococcioni and Stefano de Gironcoli [Phys. Rev. B 71, 035105 (2005)] as implemented in VASP. By using this scheme, we calculated the U value for the d -states of different Lu atoms (near and far N): The results are presented in fig. R5, showcasing a calculated U value of 1.60 eV for Lu

FIG. 5. In both graphs are represented the occupation number of d -orbitals as a function of the additional strength (in eV) of the spherical potential acting on the d -manifold for the self-consistent and the non self-consistent response, in the $\text{LuH}_{0.875}\text{N}_{0.125}$ system with two H_2 molecules. The values of the Hubbard term are computed for the Lu atom near N, with $U = 1.6$ eV (a), and far from N, with $U = 1.95$ eV (b).

atoms in close proximity to N and a U value of 1.95 eV for Lu atoms situated farther away from N. Accounting for the new value of the Hubbard term we report in Fig. R6 the density of states of our representative unit cell with relax of internal parameters, including d -states in the valence corrected with $U = 1.6$ eV compared with the same calculation without the Hubbard correction.

The comparison between the DOS at the Fermi level clearly show that the region around the Fermi level is, even in this case, qualitatively unaffected, meaning that the Hubbard correction does not alter the results and conclusions of our study. This is confirmed for example by the calculation of the total energy difference between our new proposed "molecular" phase and the previous considered $\text{LuN}_{0.125}\text{H}_{2.875}$ phase, which results in line with what already calculated without inclusion of Hubbard correction of on f - and d -states. Additionally, structural and electronic parameters of the molecular hydrogen are robust with respect to the computational setup, as shown in Table I. The Bader charge analysis shows practically no change in the excess charge populating the H_2 orbitals, which results in a constant H-H bond length. Thus, despite the interesting suggestion from the Referee, the over-delocalization of electrons of Lu atoms due to correlation errors does not impact

FIG. 6. Comparison between the DOS for the $\text{LuH}_{0.875}\text{N}_{0.125}$ system with two H_2 molecules with and without the Hubbard correction on d states ($U_d = 1.6$ eV) and with the f -states in core.

U (eV)	Bader charge for H_2 bond	
	H in H_2 (e)	length (\AA)
0	1.08	0.84
1.6	1.08	0.84
4	1.10	0.83
6	1.11	0.83

TABLE I. **Influence of the U parameters.** Bader charge of H in molecular form, and H_2 bond length, as obtained for the N doped system (N in tetragonal site, and two H_2 molecules) by using different values of U correcting the electronic correlation on Lu- d orbitals.

the properties of molecular hydrogen to a sizable extent.

We can safely conclude that our computational approach correctly describes the chemical and physical properties of the Lu-N-H compound, even against the inclusion of additional contributions to electronic correlations. However, we recognize the importance of a careful characterization of our results in term of the corrections to the electronic correlation:

Therefore, we have added a statement in the Methods and a Section in the Supplementary

Materials.

Point: 2

There is a lack of compelling evidence to support the notion that the formation of H₂ molecules enhances the superconducting transition temperature (T_c). Figure 2 does not indicate any discernible impact of H₂ on the electronic structure around the Fermi level, while Figure 3 suggests that the H₂ peaks make no substantial contribution to the overall electron-phonon coupling. Consequently, the T_c observed does not exceed that of numerous Lu-N-H systems lacking H₂ units, indicating that the presence of H₂ does not lead to a notable increase in T_c.

Answer:

We disagree with the Referee's comment, since H₂ molecules show a key role in the onset of the superconductive phase.

In fact, the role of molecular hydrogen is manifold. On one hand, they dictate the electronic properties at Fermi. The two Lu-derived flat bands and van Hove singularities close to the Fermi level (Fig. 2) are indeed promoted by the formation of H₂ molecules (as discussed in the main text). We note that van Hove singularities are not present in the pristine compound with no molecule, as reported in several publications [*e.g.*, Nat. Comm. 14, 5367 (2023)]. In N-doped LuH₃, in the absence of molecules, or if these do not exceed the number of impurities N, the system is insulating, so it cannot give rise to the superconductive state.

Additionally, the H₂ molecules make the (superconducting) structure dynamically stable at the harmonic level and at zero pressure. Conversely, in these conditions, the originally proposed Fm-3m phase of LuN_{0.125}H_{2.875} shows imaginary phonons (as demonstrated by the plethora of theoretical works appeared in literature in last months, *e.g.* Phys. Rev. Res. 5, 043238 (2023)).

Finally, we have also analyzed the direct contributions of molecular hydrogen to the overall electron-phonon coupling. This analysis is reported as reply to referee A, point 5, and in Section VII in the Supplementary Materials. In brief, we show that the molecules account for 25% of the overall electron-phonon coupling constant contribution.

Therefore, our calculations show undoubtedly the role of the molecular phase in stabilizing the (zero pressure, low T_c) superconductivity in LuN_{0.125}H_{2.875}.

We believe that the additional information included in the revised version of the manuscript and SM clarify this aspect.

REVIEWER COMMENTS

Reviewer #1 (Remarks to the Author):

The authors have satisfactorily addressed my queries. I recommend the manuscript for publication.

Reviewer #2 (Remarks to the Author):

I am satisfied with the revised manuscript and I think it should be accepted for publication.

Reviewer #3 (Remarks to the Author):

The main point of the manuscript is that N-doping stabilizes the hydrogen in a molecular phase and leads to the emergence of 0 pressure and low-temperature superconductivity. It's indeed very interesting to have the sight on potential H₂ molecules, as the formation and effects on superconductivity are not commonly discovered and discussed in other 0 K calculations. However, to convince the readers that such systems could potentially be accessible in experiments, the following evidence will be necessary: a) the possibility of forming the H₂ and b) the influence of H₂ units on the electronic and electron-phonon interaction behavior. The current manuscript does not provide enough information about these points and therefore, I cannot recommend it for publication in Nature Communications.

Specific Points:

1. Regarding “the possibility of forming H₂ molecules”: the current evidence on the possibility of H₂ formation is from the molecular dynamics simulation, where the H₂ molecules were observed in equilibrium trajectories at finite temperature. This could result in either a crystal, or a superionic phase where hydrogen sublattice is in “dynamic equilibrium”. For BCS superconductivity, a long-range periodicity is a must, and it must be proved that a real crystal could be obtained within the temperature range. The current manuscript shows the statistical results of bond lengths, energies, volumes, as well as the snapshot of the trajectory, which does not tell anything on the dynamic stability at finite temperature. It is suggested to discuss the temperature at which the structure or sublattice containing H₂ units may start to melt. Providing this information would be valuable for future studies and enhance the manuscript's appeal to researchers relying on high-impact factor journals.

2. Beyond the possibility of dynamic stability at finite temperatures, it's also important to discuss the possibility from a thermodynamic point of view. Free energy, including formation enthalpy at 0 K, as well as the free energy at finite temperature, should be provided to justify this “possibility”. The reader should be convinced: why is it possible to form this structure rather than phase separation into NH₃, LuH₂, et al.? Regarding the comments “molecular hydrogen in the thermodynamical stabilization of the system”, the evidence has been shown is on the 0 K dynamic stability rather than thermodynamics.

3. Regarding the comment “and may indicate unique conditions to drive the emergence of a superconducting phase”, in the analysis the forming H₂ will create insulators and metallic states. Since H₂ does not participate in the formation of frontier orbitals at all, the flat bands and vHs are formed by shifting of the Fermi level. This is not a unique privilege of H₂ units. Defect stoichiometry as Lu₈H₂₁N₁ will form insulator, and Lu₈H₁₉N₁ will have the same electronic structure as in the superconducting one, and there's no proposed structures on these stoichiometries might not be because they don't exist, but because people didn't explore these stoichiometries thoroughly. It should be stated as unique unless it's proved to be inaccessible otherwise.

4. Regarding Fig. SF12, the eigenvector in these modes shows collective motion of both H₂ and H⁻, it's hard to tell whether H₂ really contributes to the T_c or its from purely N/H⁻. There is a possible way to confirm the contribution here: as it shows, electron-phonon coupling changes the shape of Lu bands at E_f by modifying its electronic structures. Figure d, e, and

f all shows that the displacements lead to electrons filling the hole pocket of Lu near the Gamma point, resulting in a downward shifting of eigenstates. If these electrons are from the H2: occupation number of H2 will get reduced, result in a changing of Bader charges or a negative charge density difference (CDD). Tracking the Bader charge or CDD on H2 along the displacement will tell us if the argument is true. By incorporating these aspects into the analysis, a more comprehensive and compelling argument for the contribution of H2 towards superconductivity is provided, aligning with the expectations of a Nature Communications publication.

Responses to Referees' comments for the paper
***"Evidence of Molecular Hydrogen in the N-doped LuH₃ System:
a Possible Path to Superconductivity?"***

Cesare Tresca,^{1,*} Pietro Maria Forcella,² Andrea Angeletti,^{3,4} Luigi
Ranalli,^{3,4} Cesare Franchini,^{4,5} Michele Reticcioli,^{4,†} and Gianni Profeta^{2,1}

¹*CNR-SPIN c/o Dipartimento di Scienze Fisiche e Chimiche,
Università degli Studi dell'Aquila, Via Vetoio 10, I-67100 L'Aquila, Italy*

²*Dipartimento di Scienze Fisiche e Chimiche,
Università degli Studi dell'Aquila, Via Vetoio 10, I-67100 L'Aquila, Italy*

³*University of Vienna, Vienna Doctoral School in Physics,
Boltzmannngasse 5, 1090 Vienna, Austria*

⁴*University of Vienna, Faculty of Physics and Center for Computational Materials Science,
Kolingasse 14-16, 1090 Vienna, Austria*

⁵*Dipartimento di Fisica e Astronomia, Università di Bologna,
Viale Berti Pichat 6/2, I-40127 Bologna, Italy.*

* cesare.tresca@spin.cnr.it

† michele.reticcioli@univie.ac.at

Point-by-point replies to Referees' Reports

(In the new version of the paper, changes are marked in red.)

Reviewer: A

Comment:

The authors have satisfactorily addressed my queries. I recommend the manuscript for publication.

Answer:

We thank the Referee for her/his queries helping us to improve the quality of our manuscript.

Reviewer: B

Comment:

I am satisfied with the revised manuscript and I think it should be accepted for publication.

Answer:

We thank the Referee for her/his stimulating comments and its positive judgement.

Reviewer: D

Comment:

The main point of the manuscript is that N-doping stabilizes the hydrogen in a molecular phase and leads to the emergence of 0 pressure and low-temperature superconductivity. It's indeed very interesting to have the sight on potential H₂ molecules, as the formation and effects on superconductivity are not commonly discovered and discussed in other 0 K calculations. However, to convince the readers that such systems could potentially be accessible in experiments, the following evidence will be necessary: a) the possibility of forming the H₂ and b) the influence of H₂ units on the electronic and electron-phonon interaction behavior. The current manuscript does not provide enough information about these points and therefore, I cannot recommend it for publication in Nature Communications.

Answer:

We would like to thank the Referee for her/his comments and for underlining the importance of our work as well as for mentioning the 'very interesting' and novel aspects of studying the role of H₂ molecules on superconductivity. In the following, we provide comprehensive responses to the Referee's concerns: In particular, we show that (a) H₂ molecules are indeed formed in hydrides, as evident not only from first principles calculations but also from experiments, and that (b) H₂ has a key role in the stabilization of the superconducting phase in our calculations.

In order to clarify the aspects raised by the Referee, we have performed additional calculations and included the additional results into the manuscript, which we believe is clearer in many aspects (see below for the detailed discussion of every single point). We thank

the Referee as we think that the inclusion of the answer of the suggested points, in the new version of the manuscript, have increased its quality, we believe that the amended manuscript could be positively considered for publication in Nature Communications.

Point: 1

Regarding “the possibility of forming H₂ molecules”: the current evidence on the possibility of H₂ formation is from the molecular dynamics simulation, where the H₂ molecules were observed in equilibrium trajectories at finite temperature. This could result in either a crystal, or a superionic phase where hydrogen sublattice is in “dynamic equilibrium”. For BCS superconductivity, a long-range periodicity is a must, and it must be proved that a real crystal could be obtained within the temperature range. The current manuscript shows the statistical results of bond lengths, energies, volumes, as well as the snapshot of the trajectory, which does not tell anything on the dynamic stability at finite temperature. It is suggested to discuss the temperature at which the structure or sublattice containing H₂ units may start to melt. Providing this information would be valuable for future studies and enhance the manuscript’s appeal to researchers relying on high-impact factor journals.

Answer:

We thank the Referee for the insightful observations and comments. The formation of the H₂ molecule within the N-doped LuH₃ system is supported by several evidences already discussed in the manuscript. In fact, molecular dynamics simulations indicate the spontaneous formation of these molecules already at low temperatures (0 to 10 K). These molecules remain stable in our calculations up to temperature as high as 300 K.

This finding is also confirmed by total energy calculations performed on different configurations, which corroborate the hypothesis that the system with molecular hydrogen is the ground state (as discussed in the main text and also in the reply to the following question). Furthermore, harmonic phonon dispersion analysis confirms the dynamical stability of our system at 0 K: In the presence of H₂ molecules, all phonon modes are real, while the molecular-free phase shows imaginary modes.

Moreover, recent experimental works show signature of H₂ molecules in the Raman spectra *e.g.*, see the retracted Nature volume 615, pages 244–250 (2023), Nature 620, 72 (2023), Phys. Rev. B 108, 214505 (2023), and the recent publication Matter Radiat. Extremes 9, 037401 (2024) that we have now included as reference in the main text.

FIG. 1. Pair correlation functions at 100, 200 and 300 K in panel a, b, c, respectively, for H-H, Lu-H, Lu-Lu pairs.

adopted: In particular, the Langevin thermostat and friction coefficient of our NpT calculations affect the quantitative aspects of this analysis. In order to properly evaluate the presence of a superionic behavior, molecular dynamics simulations in different framework would be required, which is out of our scope. This analysis substantiates the stability of the proposed structure.

We note also that, upon formation of H_2 molecular units, some of the H atoms involved in the process present a temporary diffusive behavior. Figure 2 shows the time evolution of selected hydrogen atoms. Atoms not involved in the molecule formation vibrate around their equilibrium site (see atom H^0); conversely, some of the atoms forming molecules move away their original position during the formation process. This might possibly suggests

As suggested by the Referee, we also investigated the mobility of H atoms in the crystal. Figure 1 shows the pair correlation functions as obtained for the molecular dynamics simulation at 100, 200 and 300 K. The sharp peaks indicate strong correlations at specific distances, also matching the material’s crystal lattice parameters (as well as the H_2 bond length for molecular hydrogen, marked in the Figure). These peaks broaden and diminish in intensity with increasing distance and increasing temperature: Especially at 300 K, in particular for the disordered H atoms (sitting on/around lattice sites or forming molecules), the peaks appear seemingly broad but remain visible up to large distance. Therefore we conclude that there is no clear evidence of the diffusive liquid-like pattern associated with the mobility of ions in a superionic phase; however, we note that this kind of analysis is sensitive to the computational framework

FIG. 2. Trajectory of selected hydrogen atoms highlighted in fuchsia (in their initial position), at every 10 fs interval, during 0.3 ns long simulation at 300 K. The atom H^0 representative of the almost totality of the hydrogen atoms, only oscillates around its fixed sites. H atoms involved in molecular bonds may sporadically diffuse away from their stationary site (H^1 – H^4). The gradient color of the trajectories indicates the time direction, blue for initial positions and red for the final ones.

an increased mobility that can contribute to a ionic conductivity. We observe a similar behavior at all temperature explored, upon formation and breaking of molecular bonds. However, as mentioned for the pair correlation function, these results, are highly affected by the Langevin thermostat and friction coefficient adopted in the NpT calculations. We also note that the structural integrity of the Lu-hosting matrix remains unaffected by any potential hydrogen disorder, maintaining an ordered face-centered cubic (fcc) lattice. This observation underscores the robustness of our system.

Exploring the behavior of our system at temperatures above room temperature and determining the melting temperature of the H-sublattice or a potential superionic phase could indeed be intriguing, and potentially associated to a resistivity drop also in relationship to recent findings [Matter Radiat. Extremes 9, 037401 (2024)]. However, we believe that such endeavors fall outside the scope of the current study: The aim of this work is to provide solid evidence for the stabilization of a low-temperature superconducting phase due to H₂ molecule formation in N-doped LuH₃. Establishing a superionic regime would require the computation of the mean square displacement and diffusion coefficient using an NVE ensemble, *i.e.*, a completely different computational framework which requires a considerable amount of additional calculations. These topics warrant a dedicated investigation beyond the focus of our present work.

Following the suggestions from the Referees, we have included the description of the pair correlation functions as well as the trajectory evolution of H atoms in the SM.

Point: 2

Beyond the possibility of dynamic stability at finite temperatures, it's also important to discuss the possibility from a thermodynamic point of view. Free energy, including formation enthalpy at 0 K, as well as the free energy at finite temperature, should be provided to justify this "possibility". The reader should be convinced: why is it possible to form this structure rather than phase separation into NH₃, LuH₂, et al.? Regarding the comments "molecular hydrogen in the thermodynamical stabilization of the system", the evidence has been shown is on the 0 K dynamic stability rather than thermodynamics.

Answer:

We agree with the Referee that studying the thermodynamic stability of the molecular phase is, in principle, an interesting aspect.

The investigation of the stability of disordered phases requires an immense amount of work, which goes beyond our scope. Our goal is to show solid arguments for a proof-of-concept study aiming to propose that H₂ molecules can form in hydrates, and impact on the electronic properties of the systems (*e.g.*, possibly leading to the stabilization of a superconducting phase).

In the literature, the phase-diagram analysis has been already reported for ordered phases of the N-doped LuH_x ternary compound [see for example: Phys. Rev. B 108, L020102 (2023), Phys. Rev. B 108, 014511, Phys. Rev. Materials 8, L021801, Nature Communications (2023)14:5367, Nature Communications (2024)15: 441 (2024), arXiv:2307.10699]. The structural searches are usually performed using smaller unit cells with respect to the one used in our simulations containing molecular hydrogen, making difficult to compare the stoichiometries. The construction of a new convex hull using our stoichiometries (and the larger unit cell) will require a large amount of calculations and dedicated efforts (as demonstrated by the numerous works focusing exclusively on this very specific topic) which is clearly not the aim of our work.

Anyway, the results available in literature (see the references cited above) for the ordered N-doped LuH₃ with Lu-fcc lattice and a stoichiometry close to our system, falls in few meV/atom above the convex-hull as calculated considering possible phase separation into

the constituents NH_3 , LuH_2 et al.. It is important to emphasise that even thermodynamically stable phases are not always experimentally realized, in favour of metastable phases. In fact, even in the literature mentioned above, the thermodynamically stable phases predicted in the calculations are in contrast with XRD spectra measured by different groups for LuH_x [Nature 615, 244 (2023), Nature 620, 72 (2023), AIP Advances 13, 065117 (2023), Phys. Rev. B 108, 214505 (2023)]. Such deviations from the hull can be considered as an acceptable range for real experimental realization for hydrides.

In our case, having discovered new phases, containing hydrogen molecules, with lower total energies with respect to the high-symmetric phase ($\simeq 10$ meV/atom, as discussed into the manuscript) we are further increasing the stability of the compound, probably even below the convex-hull. This finding clearly proves that the molecular phase is the ground state of the system, although a complete exploration of the phase-diagram is hindered by the complexity of the problem.

Motivated by the Referee’s comment, we decided to further investigate possible spurious pressure effects on the energetic of the considered systems. As discussed in the Methods, in our calculations (except for the molecular dynamics), we model all systems using the experimental value for the lattice parameters. Selecting the experimental lattice constant is the most natural choice to build a representative model for an intrinsically disordered system. Obviously, this leads to a low residual virtual pressure ($P \sim 40 \div 80$ kbar). In order to correctly evaluate the enthalpy of our model representative of the molecular hydrogen phase, we fully relaxed the structure to reach the computational $P^{DFT}=0$ kbar (which results in slight deformations of the cubic lattice, ending in a triclinic phase, with an average expansion of about 3% of the lattice constants, with respect the experimental one). This slight deformation practically does not affect the stability of the molecules nor the electronic properties with respect to the results already reported in the main text for the system at the experimental lattice constant. In particular, the calculated enthalpies of the various models (0, 1 and 2 molecules) considering structural relaxation at zero pressure, are in agreement with the total energies already found for the same models at fixed experimental volume: with respect to the system without H_2 molecules, the formation enthalpy for the system with one H_2 molecule is $\Delta H = -0.010$ eV/atom and for two H_2 molecules is $\Delta H = +0.056$ eV/atom.

To check the effect of cell relaxation on the electronic and dynamical properties, we report in Fig.3 the electronic band structures, the dynamical and Eliashberg function of the

FIG. 3. Electronic dispersions and density of states (left) and dynamical properties and Eliashberg Function (right) of the $\text{LuH}_{2.875}\text{N}_{0.125}$ with two molecules structural relaxed up to theoretical $P=0$ kbar pressure.

theoretical $P^{DFT}=0$ kbar system with 2 H_2 molecules. As already mentioned, the band structure is practically unaffected, showing a metallic, low dispersive band at the Fermi level. The phonon band structure shows a overall dynamically stable phase which underlines the crucial role the molecules play in the stabilization of the system. For example, if we consider the proposed Fm-3m phase, without molecules, the system shows a severe dynamical instability with large imaginary ($i400 \text{ cm}^{-1}$) phonons extending over the whole Brillouin zone. While, the inclusion of molecules make most of the phonons dynamically stable with the only exception of a small instability limited around the L^* point of the Brillouin Zone ($\omega(L^*) \sim i60 \text{ cm}^{-1}$). This is a somehow expected result: a long-range distortion is indeed expected for a model-system which has a smaller unit cell than the one used for MD simulations, where we found the spontaneous formation of molecules, signaling the presence of (at least) orientational disorder of the hydrogen molecules in the system.

Finally, we also highlight that the phonon-DOS and, more importantly, the α^2F curve are practically indistinguishable from those obtained using experimental parameters, presented in the main text. The estimation of T_c for the relaxed $P^{DFT}=0$ kbar system (arbitrarily neglecting the contribution of the imaginary modes), is also in line with the estimation reported in the article (with $T_c \sim 20 \text{ K}$).

We included a discussion of these aspects also in the Methods Section and in the SM.

Point: 3

Regarding the comment “and may indicate unique conditions to drive the emergence of a superconducting phase”, in the analysis the forming H₂ will create insulators and metallic states. Since H₂ does not participate in the formation of frontier orbitals at all, the flat bands and vHs are formed by shifting of the Fermi level. This is not a unique privilege of H₂ units. Defect stoichiometry as Lu₈H₂₁N₁ will form insulator, and Lu₈H₁₉N₁ will have the same electronic structure as in the superconducting one, and there’s no proposed structures on these stoichiometries might not be because they don’t exist, but because people didn’t explore these stoichiometries thoroughly. It should be stated as unique unless it’s proved to be inaccessible otherwise.

Answer:

We agree with the referee that other superconducting phases might, in principle, arise in compounds with different stoichiometry (although systematic studies on out-of-stoichiometry compounds have failed so far to find superconductivity at low pressure). We agree that the expression “unique conditions” deceives the reader from the intended message: In fact, we meant that the H₂ molecules show a special/key role (rather than unique) for the dynamical stabilization of the system.

Our proof-of-concept study shows strong evidences for the stabilization of a LuH_{2.875}N_{0.125} phase containing hydrogen in molecular form that can possibly host a (low-temperature, zero-pressure) superconducting phase.

As demonstrated by the band structure calculations for our compound, the appearance of flat bands around the Fermi level is a consequence of the H₂ formation and these electronic states will eventually couple with Lu- and H₂-derived modes (see SM).

We cannot obviously exclude that different stoichiometries could lead to similar results.

The study of different stoichiometries of Lu-hydride (and possibly other) within our computational approach will surely represents an interesting follow-up research that could possibly bring to interesting superconducting phases. We believe that our paper proposes a new computational paradigm which could inspire several works looking for superconducting hydrides.

We thank the referee for the comment, and we have corrected the sentence in the amended

version of the main text.

Point: 4

Regrading Fig. SF12, the eigenvector in these modes shows collective motion of both H₂ and H-, it's hard to tell whether H₂ really contributes to the T_c or its from purely N/H-. There is a possible way to confirm the contribution here: as it shows, electron-phonon coupling changes the shape of Lu bands at E_f by modifying its electronic structures. Figure d, e, and f all shows that the displacements lead to electrons filling the hole pocket of Lu near the Gamma point, resulting in a downward shifting of eigenstates. If these electrons are from the H₂: occupation number of H₂ will get reduced, result in a changing of Bader charges or a negative charge density difference (CDD). Tracking the Bader charge or CDD on H₂ along the displacement will tell us if the argument is true. By incorporating these aspects into the analysis, a more comprehensive and compelling argument for the contribution of H₂ towards superconductivity is provided, aligning with the expectations of a Nature Communications publication.

Answer:

We thank the Referee for stimulating us in a deeper analysis of this aspect. As already discussed, the presence of molecular hydrogen is key for the dynamical stabilization of the structure at ambient pressure. This structure, stabilized by the H₂ molecules, also turns out to develop a low-temperature superconductive phase. In the following, we unveil the contribution of H₂ (and H) modes to the electron-phonon coupling.

A first evidence about the contribution of H₂ and H to the determination of the superconductive state can be found in Figure SF12. Here, we show the band structure as a function of the atomic displacement along some selected most-coupled phonon modes. The effect of H₂ (and H) derived phonons on the deformation potential, and thus on the electron-phonon coupling is evident: a relevant modulation of the electronic band structure at the Fermi level. In addition, in the manuscript, we report the projected $\alpha^2 F(\omega)$ on different species, showing a clear and relevant contribution from H-, H₂ and N- derived modes. Anyway, to look at the same information from a different perspective, we have gladly accepted the Referee's suggestion and have calculated the Bader charges of the constituents as a function of the atomic displacements along the 3 modes considered in Fig.SF12 (results are shown in

FIG. 4. We report the Bader charge analysis as obtained from VASP calculations as a function of the atomic displacements along the phonon eigenvectors we are interested in. The average value of the Bader charge is reported for every element (in units of the electronic charge, e).

Fig.4).

Activating the atomic oscillation (we considered only positive displacement), we find a clear charge transfer mainly involving H-, H₂ and Lu atoms, which show an appreciable relative variation ($\sim 1.9\%$, $\sim 1.8\%$ and $\sim 0.7\%$, respectively). In particular, for the analyzed

displacements, H⁻ acts as donor, while H₂ and Lu as acceptors. In particular, the most coupled modes among the three analysed (27 and 40) give larger (and similar) charge transfer to the H₂; instead the less coupled mode (29) produces a smaller variation in the Bader charge of molecules. This aspect can be viewed as the real-space manifestation of the role of H₂ in determining the electron-phonon coupling of the representative modes 27 and 40.

We add a comment in the new version of the SM.

REVIEWER COMMENTS

Reviewer #1 (Remarks to the Author):

I am very happy with the author's response and recommend it for publication.

Reviewer #2 (Remarks to the Author):

I am satisfied with the revised manuscript and I think it should be accepted for publication.

Reviewer #4 (Remarks to the Author):

The authors have addressed all the issues and comments raised by the Referees in a very detailed and thorough manner, which is appreciated. I also think that the information presented in the manuscript definitely deserves to be published, but there are still a few important points that need to be revised or addressed before I can see myself able to recommend publication.

1.) As has been suspected and now confirmed (<https://www.nature.com/articles/d41586-024-00976-y>) much of the results of the now retracted Nature paper are based on fraudulent conduct. In light of these circumstances I think the authors should revise the wording when referring to that paper throughout the text and that this retracted paper should not be used to compare or validate the computational work. In fact, there are by now many other experimental works that are a lot more trustworthy and can be used as reference. I understand that this is a unique situation and that the manuscript has been originally submitted before all the conclusive evidence against the Rochester group has been made public, but the authors could use this as an opportunity to revise their manuscript accordingly. I suggest to move the focus of the manuscript even stronger from

"trying to find the structure reported in the (retracted) Nature paper" to "reporting a new and interesting phase of the Lu-H-N system". On a side note, the Rochester paper has not been retracted because all attempts to reproduce the results have been in vain, as stated in the introduction of the manuscript, but because several authors of that paper asked for the retraction due to doubts about the reported data and its origin.

2.) Relating to the first point, I was a bit surprised to see that very little attempts have been made to compare the computational results to trustworthy experimental data. For example, XRD patterns of the proposed simulated structure(s) could be compared to actual XRD measurements as reported in many works on the Lu-N-H system since, as has been done in many other theoretical works on this topic already. Also, I think it would be important to have a good comparison between the simulated Raman data to experimental ones, maybe with a figure or a table, and not just in 2 sentences. Volume-vs-pressure curves would also be interesting to compare, etc.

3.) I also want to note that no experimental work I'm familiar with found any superconducting transition above 2K below pressures of 50GPa in the Lu-N-H system, while in this work a sizeable T_c of about 13K is predicted. If the structure proposed by the authors is indeed the one from experiment(s), why has no superconducting phase transition been observed so far experimentally?

4.) I find the wording in the manuscript odd at some places. For example, I'm not sure I understand what the authors want to say in the last paragraph of the Conclusion. It is very convoluted and to me does not provide any concrete information. Also, what exactly are the authors trying to say with "These features are not present in the proposed (molecule free) LuH-N phases appeared in the recent literature, and may indicate key conditions to drive the emergence of a superconducting phase."? The authors need to be more on point and convey their thoughts here more clearly.

5.) I don't think the sentence "As an illustration, numerous studies have delved into the LuHN ternary hydride, forecasting relatively high-temperature superconducting phases (up to 100K), yet these predictions have not been experimentally confirmed to date" is correct, in particular with respect to the given references. Out of all the 6 works cited in this context, only one (Phys. Rev. Materials 8, L021801) actually predicts a structure with T_c around or

above 100K, all other works clearly state that no high-T_c conventional superconductivity can be found in any of the considered phases. Incidentally, the paper mentioned above is authored by the Hemley group who also are the only ones that claim to have been able to reproduce the Rochester results experimentally. Given the fact that the Hemley and Rochester group have authored papers together before and that R. Hemley has also been involved in the investigation of fraud of the University of Rochester as a reviewer, I remain sceptical about these works.

6.) In Fig. SF7 the authors give an example of the DOS for a representative metallic configuration extracted from their MD runs, but to me this looks more like the phase having a band gap with the Fermi level touching the bands due to electronic smearing or similar, i.e., more of a numerical artifact. It is definitely not an example of a textbook metallic DOS. I'd appreciate if the authors could comment on that and why they chose this figure.

7.) Many of the cited arXiv papers have since been published in various journals and the references should be updated.

To summarize, I do think that this is a nice and interesting work that is potentially interesting to a broader audience, but in my opinion, the manuscript needs to be revised to better convey the actual research results and to de-couple more strongly from the retracted Rochester works. I'm also a bit hesitant whether Nature Communications is the best journal for this work, but acknowledge that a revised and well-worded/presented version could very well be.

Responses to Referees' comments for the paper
***"Evidence of Molecular Hydrogen in the N-doped LuH₃ System:
a Possible Path to Superconductivity?"***

Cesare Tresca,^{1,*} Pietro Maria Forcella,² Andrea Angeletti,^{3,4} Luigi
Ranalli,^{3,4} Cesare Franchini,^{4,5} Michele Reticcioli,^{4,†} and Gianni Profeta^{2,1}

¹*CNR-SPIN c/o Dipartimento di Scienze Fisiche e Chimiche,
Università degli Studi dell'Aquila, Via Vetoio 10, I-67100 L'Aquila, Italy*

²*Dipartimento di Scienze Fisiche e Chimiche,
Università degli Studi dell'Aquila, Via Vetoio 10, I-67100 L'Aquila, Italy*

³*University of Vienna, Vienna Doctoral School in Physics,
Boltzmannngasse 5, 1090 Vienna, Austria*

⁴*University of Vienna, Faculty of Physics and Center for Computational Materials Science,
Kolingasse 14-16, 1090 Vienna, Austria*

⁵*Dipartimento di Fisica e Astronomia, Università di Bologna,
Viale Berti Pichat 6/2, I-40127 Bologna, Italy.*

* cesare.tresca@spin.cnr.it

† michele.reticcioli@univie.ac.at

Point-by-point replies to Referees' Reports

(In the new version of the paper, changes are marked in red.)

Reviewer: 1

Comment:

I am very happy with the author's response and recommend it for publication.

Answer:

We thank the Referee for her/his suggestions and final positive evaluation.

Reviewer: 2

Comment:

I am satisfied with the revised manuscript and I think it should be accepted for publication.

Answer:

We would like to thank the Referee and we are pleased to have met her/his requests.

Reviewer: 4

Comment:

The authors have addressed all the issues and comments raised by the Referees in a very detailed and thorough manner, which is appreciated. I also think that the information presented in the manuscript definitely deserves to be published, but there are still a few important points that need to be revised or addressed before I can see myself able to recommend publication.

Answer:

We are very happy about the Referee's positive opinion of our work, and that she/he recognizes and appreciates our efforts in responding to previous reviews.

Below we list our detailed responses to the reported criticisms.

Point: 1

As has been suspected and now confirmed (<https://www.nature.com/articles/d41586-024-00976-y>) much of the results of the now retracted Nature paper are based on fraudulent conduct. In light of these circumstances I think the authors should revise the wording when referring to that paper throughout the text and that this retracted paper should not be used to compare or validate the computational work. In fact, there are by now many other experimental works that are a lot more trustworthy and can be used as reference. I understand that this is a unique situation and that the manuscript has been originally submitted before all the conclusive evidence against the Rochester group has been made public, but the authors could use this as an opportunity to revise their manuscript accordingly. I suggest to move the focus of the manuscript even stronger from "trying to find the structure reported in the (retracted) Nature paper" to "reporting a new and interesting phase of the Lu-H-N system". On a side note, the Rochester paper has not been retracted because all attempts to reproduce the results have been in vain, as stated in the introduction of the manuscript, but because several authors of that paper asked for the retraction due to doubts about the reported data and its origin.

Answer:

We agree with the Referee, thus, we gladly take the opportunity to revise all statements referring to the now retracted Nature paper, and present our results in a different perspective. We want to underline further that our work is not linked or based at all on the results contained on the retracted paper, but, being a theoretical and computational study, sheds light on new phenomena which could be active on different hydride compounds. We make this aspect clear in the new version of the manuscript.

Point: 2

Relating to the first point, I was a bit surprised to see that very little attempts have been made to compare the computational results to trustworthy experimental data. For example, XRD patterns of the proposed simulated structure(s) could be compared to actual XRD measurements as reported in many works on the Lu-N-H system since, as has been done in many other theoretical works on this topic already. Also, I think it would be important to have a good comparison between the simulated Raman data to experimental ones, maybe

with a figure or a table, and not just in 2 sentences. Volume-vs-pressure curves would also be interesting to compare, etc.

Answer:

As described in the manuscript, several and recent experimental XRD spectra show that the Lu-hydride system is characterized by the Lu-*fcc* crystal structure. Accordingly, our molecular dynamics simulations retain such Lu-*fcc* lattice. We thank the Referee for her/his suggestion, which has motivated us to include in the Supplementary Materials a comparison between experimental XRD data and simulated spectra from our MD-relaxed structures: To this aim, we added a new figure in the Supplementary Material (Sec.X in SM) showing a satisfactory agreement between simulated and experimental spectra.

Additionally, as suggested by the Referee, we include in the revised version a comparison between our theoretically-predicted Raman active frequencies with recent experimental Raman results (not from the Rochester group), showing an overall good agreement. It is important to mention that the only Raman spectra available for the very high frequencies for the *fcc* crystal ($> 2000\text{cm}^{-1}$) are those reported by the Rochester group, which we omit to report, although in very good agreement with our proposed structural models. The only high frequency Raman data are those of the Lu-N-H ternary system in the (nominal) trigonal phase[Matter Radiat. Extremes 9, 037401 (2024)]. Anyway, as the trigonal structure is a deformation of the *fcc*-cubic lattice[Crystalline Materials 222, 580–594 (2007)], a comparison, even qualitative, with the Raman spectrum of the trigonal phase can be legitimate and interesting. Given the lack of data for the *fcc*-crystal, we call for future experiments to address this aspect, *i.e.*, experiments tackling the detection of molecular hydrogen in hydrides. We modified the manuscript accordingly.

Finally, regarding the last point mentioned by the Referee (“Volume-vs-pressure curves would also be interesting to compare”), we agree about the potential interest of the analysis of volume-vs-pressure curves; however, since our work is intentionally focused on near ambient-pressure conditions, we leave such additional analysis for future works. We comment about this point the Conclusions: “*The emergence of a low-temperature superconductivity driven by H_2 molecules stabilized by N impurities could also stimulate further theoretical studies inspecting the role of pressure, local dis-homogeneity of H , and/or different amount/type of doping with respect to the stability of the molecular phase*”

Point: 3

I also want to note that no experimental work I'm familiar with found any superconducting transition above 2K below pressures of 50GPa in the Lu-N-H system, while in this work a sizeable T_c of about 13K is predicted. If the structure proposed by the authors is indeed the one from experiment(s), why has no superconducting phase transition been observed so far experimentally?

Answer:

We thank the Referee for giving us the opportunity to discuss this very important and crucial point. The discrepancy between theoretical predictions of the superconducting critical temperature and the measured values is a very debated issue in the field of superconducting hydrides. In fact, during the last 4-5 years several predictions of new superconducting hydrides have been published (even showing extraordinary critical temperatures), yet no experimental confirmation has been so far reported: This issue was discussed in detail in review and perspective papers, such as Ref. Physics Reports 856, 1 (2020), Journal of Physics Condensed Matter 34, 184, 183002 (2022).

Analogously, as also stated in the manuscript, several theoretical works have proposed sizable critical temperatures for the Lu-H-N system (at high pressure), with no confirmation from the experiments.

We believe that this discrepancy is due to the complexity of the hydride compounds, which could easily show disorder effects, phase separations, off-stoichiometric phases, temperature effects, out-of-equilibrium conditions, etc... Our work was motivated from this consideration: Thus, in our simulations, we account for part of these effects (temperature and disorder effects) demonstrating the formation of molecular hydrogen within the metallic matrix and the fundamental role of molecules have for the dynamical stabilisation of the material. Such disordered phases are difficult to realize in real samples with precise control: In fact, recent experimental studies (not only the retracted paper) show electronic phase transition (resistivity drops) only in portion of the realized samples. In our opinion, this indicates that disorder plays a crucial role, and fine control over the synthetization process is required in order to stabilize potentially superconducting phases. Our simulations highlight the importance of such a fine characterization of the samples, and could motivate

experimental works in this direction.

We have included a brief comment in the Conclusions of the revised manuscript. We also note that this concept is already introduced in the Introduction: “*This discrepancy may be attributed to challenges in accurately accounting for real experimental conditions during crystal growth.*”

Point: 4

I find the wording in the manuscript odd at some places. For example, I’m not sure I understand what the authors want to say in the last paragraph of the Conclusion. It is very convoluted and to me does not provide any concrete information. Also, what exactly are the authors trying to say with “These features are not present in the proposed (molecule free) LuH-N phases appeared in the recent literature, and may indicate key conditions to drive the emergence of a superconducting phase.”? The authors need to be more on point and convey their thoughts here more clearly.

Answer:

As suggested by the Referee, we have modified several points of the main text, in order to clarify our statements.

Point: 5

I don’t think the sentence “As an illustration, numerous studies have delved into the LuHN ternary hydride, forecasting relatively high-temperature superconducting phases (up to 100K), yet these predictions have not been experimentally confirmed to date” is correct, in particular with respect to the given references. Out of all the 6 works cited in this context, only one (Phys. Rev. Materials 8, L021801) actually predicts a structure with Tc around or above 100K, all other works clearly state that no high-Tc conventional superconductivity can be found in any of the considered phases. Incidentally, the paper mentioned above is authored by the Hemley group who also are the only ones that claim to have been able to reproduce the Rochester results experimentally. Given the fact that the Hemley and Rochester group have authored papers together before and that R. Hemley has also been involved in the investigation of fraud of the University of Rochester as a reviewer, I remain sceptical about these works.

Answer:

We agree with the Referee. It was indeed not our intention to state that there were reliable predictions of high-temperature superconductivity for Lu-N-H system; Rather, we intended to report that some metastable (or nearly dynamically stable) structures were predicted to show large critical temperature, *e.g.*, see the estimations presented in Nature Communications 14, 5367 (2023), Nature Communications 15, 441 (2024) and recently in Communications Materials 5, 61 (2024). All these referenced works looked for an explanation of the observed superconducting phase in Lu-N-H (later retracted), by proposing different (metastable or dynamically unstable) crystal structures showing sizable critical temperatures.

We reformulate the sentence to make this point clearer in the revised version of the manuscript.

Regarding Hemley's work [Phys. Rev. Materials 8, L021801], we feel that presently it can be cited in a list of references which analyze different hypothetical structures in order to find possible superconducting phases.

Point: 6

In Fig. SF7 the authors give an example of the DOS for a representative metallic configuration extracted from their MD runs, but to me this looks more like the phase having a band gap with the Fermi level touching the bands due to electronic smearing or similar, i.e., more of a numerical artifact. It is definitely not an example of a textbook metallic DOS. I'd appreciate if the authors could comment on that and why they chose this figure.

Answer:

We agree with the Referee about the importance of ensuring that the system is metallic. We carefully checked this aspect throughout the study and adopted a well converged setup; anyhow, as ultimate proof, we add now in Fig. SF7 an inset showing a detail about the band structure cutting the Fermi level around the Γ point of the Brillouin zone (which undoubtedly confirm the metallic state).

Point: 7

Many of the cited arXiv papers have since been published in various journals and the references should be updated.

Answer:

We thank the Referee, we have amended the references accordingly.

Final comment:

To summarize, I do think that this is a nice and interesting work that is potentially interesting to a broader audience, but in my opinion, the manuscript needs to be revised to better convey the actual research results and to de-couple more strongly from the retracted Rochester works. I'm also a bit hesitant whether Nature Communications is the best journal for this work, but acknowledge that a revised and well-worded/presented version could very well be.

Answer:

We thank the Referee for her/his positive opinion of our work and for all suggestions, which we found extremely useful to improve the manuscript. We hope that the work can be now positively considered for publication in "Nature Communications".

REVIEWERS' COMMENTS

Reviewer #4 (Remarks to the Author):

I appreciate the authors' efforts to revise their manuscript yet again. In my opinion, the revised manuscript is considerably improved and I recommend publication in Nature Communications.